# Byzantine-Resilient Decentralized Multi-Armed Bandits

**Jingxuan Zhu**                                                                  *jxzhu@zhejianglab.org*
*Zhejiang Lab*

**Alec Koppel**                                                                  *alec.koppel@jpmchase.com*
*J.P. Morgan AI Research*

**Alvaro Velasquez**                                                            *alvaro.velasquez@colorado.edu*
*University of Colorado Boulder*

**Ji Liu**                                                                        *ji.liu@stonybrook.edu*
*Stony Brook University*

**Reviewed on OpenReview:** *https://openreview.net/forum?id=JoYMJJdvry*

## Abstract

In decentralized cooperative multi-armed bandits (MAB), each agent observes a distinct stream of rewards, and seeks to exchange information with others to select a sequence of arms so as to minimize its regret. Agents in the cooperative setting can outperform a single agent running a MAB method such as Upper-Confidence Bound (UCB) independently. In this work, we study how to recover such salient behavior when an unknown fraction of the agents can be *Byzantine*, that is, communicate arbitrarily wrong information in the form of reward mean-estimates or confidence sets. This framework can be used to model attackers in computer networks, instigators of offensive content into recommender systems, or manipulators of financial markets. Our key contribution is the development of a fully decentralized resilient upper confidence bound (UCB) algorithm that fuses an information mixing step among agents with a truncation of inconsistent and extreme values. This truncation step enables us to establish that the performance of each normal agent is no worse than the classic single-agent UCB1 algorithm in terms of regret, and more importantly, the cumulative regret of all normal agents is strictly better than the non-cooperative case, provided that each agent has at least $3f + 1$ neighbors where $f$ is the maximum possible Byzantine agents in each agent's neighborhood. Extensions to time-varying neighbor graphs, and minimax lower bounds are further established on the achievable regret. Experiments corroborate the merits of this framework in practice.

## 1 Introduction

In multi-armed bandits (MAB) (Lattimore & Szepesvári, 2020), one is faced with the task of selecting a series of arms so as to accumulate the most reward in the long-term when rewards are incrementally revealed. The canonical performance measure is regret, which quantifies the difference in the cumulative return associated with one's chosen actions as compared with the best-in-hindsight up to some fixed time-horizon $T$. For the frequentist setting, that is, no distributional model is hypothesized underlying the reward sampling process, both lower and upper bounds on the asymptotic regret exist (Lai & Robbins, 1985). Upper confidence-bound (UCB) selection method was proposed in Auer et al. (2002a) which achieves $O(\log T)$ regret.

In this work, we focus on cooperative multi-agent extensions of this setting (Landgren et al., 2016), where each node in a directed graph now receives a distinct reward sequence and needs to select its own series of arms. Cooperative MAB has received significant attention in recent years, where agents compute locally weighted averages of their agents' arm index parameters, i.e., they execute consensus (Jadbabaie et al., 2003;

Olfati-Saber et al., 2007). Numerous works have studied this problem class in recent years for the case that agents reward distributions are homogeneous (Landgren et al., 2016; Landgrena et al., 2021; Martínez-Rubio et al., 2019; Zhu et al., 2021; Chawla et al., 2020), as well as (Lai et al., 2008; Liu & Zhao, 2010; Szorenyi et al., 2013; Kalathil et al., 2014; Bistritz & Leshem, 2018; Sankararaman et al., 2019; Wang et al., 2020; Dubey & Pentland, 2020; Shi & Shen, 2021; Madhushani & Leonard, 2020a;b; Martínez-Rubio et al., 2019), extensions to models of "collision" (Nayyar et al., 2018; Liu & Zhao, 2010; Bistritz & Leshem, 2018; Lai et al., 2008) and federated information structures (Shi & Shen, 2021; Shi et al., 2021; Zhu et al., 2021; Dubey & Pentland, 2020; Li et al., 2020; Réda et al., 2022). Most pertinent to this work are consensus-based decentralized UCB-type methods (Landgren et al., 2016; Landgrena et al., 2021; Martínez-Rubio et al., 2019; Zhu et al., 2021; Zhu & Liu, 2023). It is worth noting that in such a homogeneous setting, each agent in a multi-agent network can independently learn an optimal arm using any conventional single-agent UCB algorithm, ignoring any information received from other agents. With a careful design, decentralized multi-agent multi-armed bandits can outperform centralized single-agent upper confidence bound algorithms (Zhu & Liu, 2021; 2024).

However, it is overly narrow to consider that all agents cooperate, as many learning processes have a mixture of cooperative and competitive agents, and it is not always clear to delineate between them (Littman, 1994; Başar & Olsder, 1998). To encapsulate this dynamic, we consider that an unknown subset of agents are "Byzantine" (Lamport et al., 1982), i.e., they chose their index parameters and selects arms in an arbitrary manner and can send conflicting values to different agents. The cooperative MAB in the presence of Byzantine agents setting has intrinsic motivation in computer networks (Xiong & Jamieson, 2013; Ferdowsi et al., 2019), recommender systems (Sankararaman et al., 2019), and economics (Misra et al., 2019), to name a few, as a model of how these systems should optimize performance in the presence of uncertainty in their incentives. In particular, cooperation can respectively model filesharing, collective interest around a topic, or market stability metrics. Moreover, the concept of Byzantine agents naturally emerges when some participants seek to derail the goal of the collective, as in denial of service attacks, injection of offensive content into social media, or market manipulation. To date, however, the majority of works that study MAB in the cooperative multi-agent setting do not allow for any agents to be non-cooperative, leading to a learning process being inoperable in the presence of malefactors. Therefore, in this work, we pose the following question:

*Can decentralized multi-armed bandit algorithms outperform the classic single-agent counterpart even in the presence of Byzantine agents?*

In this work, we provide an affirmative answer to this question. Therefore, our contributions are to:

- pose the multi-armed bandit problem in the multi-agent setting in the presence of Byzantine agents;

- derive a filtering step on the parameters exchanged by agents that thresholds extreme values, which guarantees consistency in the construction of confidence sets and helps limit the deleterious impact of the Byzantine agents by yielding accurate reward mean estimates. As a result, we construct a Byzantine-resilient variant of decentralized UCB framework;

- establish that the regret performance of this method no worse than logarithmic regret bound of its single-agent counterpart (which is not true of MAB methods even in the Byzantine-free case (Landgren et al., 2016; Landgrena et al., 2021; Martínez-Rubio et al., 2019)), and is better than the non-cooperative counterpart in terms of network total regret when each agent has at least $3f + 1$ neighbors, where $f$ is an upper-bound on the number of Byzantine agents that is commonly assumed known in resilient learning (Gupta et al., 2021; Gupta & Vaidya, 2021; Kaheni et al., 2022; Liu et al., 2021b; Gupta et al., 2023). Notably, we do not require global knowledge of the network connectivity, as in Vial et al. (2021; 2022);

- experimentally demonstrate that the proposed methodology achieves lower regret compared with single-agent (non-cooperative) UCB, meaning that the sum is more than the component parts. The constructed setting is representative of a model of recommender systems, where advertising firms may share data to improve reliability of their targeted ads.

A detailed review of related work can be found in Appendix B.

## 2 Multi-agent Multi-armed Bandits

Consider a multi-agent network consisting of $N > 1$ agents, which may be mathematically formalized as a directed graph $\mathbb{G} = (\mathcal{V}, \mathcal{E})$ with $\mathcal{V} = \{1, \ldots, N\}$. Each vertex $i$ corresponds to an agent and each directed edge (or arc) defines connectivity amongst neighbors. To be more precise, we say agent $j$ is a neighbor of agent $i$ if $(j, i)$ is an arc in $\mathcal{E}$, and similarly, agent $k$ is an out-neighbor of agent $i$ if $(i, k) \in \mathcal{E}$. Each agent may receive information only from its neighbors with an inward bound arc, meaning that the directions of arcs represent the directions of information flow. We use $\mathcal{N}_i$ and $\mathcal{N}_i^-$ to respectively denote the neighbor set and out-neighbor set of agent $i$, i.e., $\mathcal{N}_i = \{j \in \mathcal{V} : (j, i) \in \mathcal{E}\}$ and $\mathcal{N}_i^- = \{k \in \mathcal{V} : (i, k) \in \mathcal{E}\}$.

All agents face the same set of $M > 1$ arms (or options) denoted as $\mathcal{M} = \{1, \ldots, M\}$. At each discrete time $t \in \{0, 1, \ldots, T\}$, agent $i \in \mathcal{V}$ must select an arm $a_i(t) \in \mathcal{M}$ from the $M$ options. If agent $i$ selects an arm $k \in \mathcal{M}$, it will receive a random reward $X_{i,k}(t)$. For any $i \in \mathcal{V}$ and $k \in \mathcal{M}$, $\{X_{i,k}(t)\}_{t=0}^T$ is an unknown independent and identically distributed (i.i.d.) stochastic process. For each arm $k \in \mathcal{M}$, all its reward variables over the network $X_{i,k}(t)$, $i \in \mathcal{V}$ share the same mean $\mu_k$ while their distributions may be different. Without loss of generality, we assume that all $X_{i,k}(t)$, $i \in \mathcal{V}$, $k \in \mathcal{M}$ have bounded support $[0, 1]$ and that $\mu_1 \geq \mu_2 \geq \cdots \geq \mu_M$, which implies that arm 1 has the largest reward mean and thus is always an optimal option.

There exist agents in the network which are able to transmit arbitrary values to their out-neighbors and capable of sending conflicting values to different neighbors strategically, which we refer to as *Byzantine agents* (Fischer et al., 1986). The set of Byzantine agents is denoted by $\mathcal{F}$ and the set of normal (non-Byzantine) agents is denoted by $\mathcal{H}$, whereby $\mathcal{V} = \mathcal{H} \cup \mathcal{F}$. Which agents are Byzantine is unknown to normal agents; however, we assume that each agent has at most $f$ Byzantine neighbors, and knows the number $f$. Knowing the number $f$ can be viewed as a limitation of this work, though it is a widely adopted assumption in the existing Byzantine-resilient algorithms in the literature.

The canonical performance measure in the multi-agent MAB setting may be formalized through the concept of *regret* among the non-Byzantine agents. That is, each normal agent $i$ seeks to select its sequence of arms $\{a_i(t)\}_{t=1}^T$ so as to minimize its contribution to a network-wide cumulative regret

$$R(T) = \sum_{i \in \mathcal{H}} R_i(T) = \sum_{i \in \mathcal{H}} \left( T\mu_1 - \sum_{t=1}^T \mathbf{E}\big[X_{a_i(t)}\big] \right), \tag{1}$$

where the sum is over all non-Byzantine agents though in the presence of Byzantine agents. Moreover, $R_i(T)$ quantifies the difference in the cumulative return $\sum_{t=1}^T \mathbf{E}\big[X_{a_i(t)}\big]$ as compared with the best-in-hindsight up to horizon $T$, which by our assumption on the ordering of $\mu_i$, is simply $T\mu_1$.

In the case when each agent does not have any neighbors, the aforementioned resilient multi-agent multi-armed bandit problem reduces to the conventional single-agent multi-armed bandit problem for each normal agent. It is well known that each normal agent can appeal to the classic UCB1 algorithm and achieve an $O(\log T)$ regret (Auer et al., 2002a). Subsequently, we develop a Byzantine-resilient decentralized multi-agent UCB1 algorithm which guarantees that all normal agents collectively outperform the scenario had they each adopted the classic single-agent UCB1 algorithm. Before continuing, we introduce an example to clarify the problem setting.

**Example 1 (Recommender Systems)** In personalized recommender systems in web services (Sankararaman et al., 2019), an application interface is charged with the task of presenting one of several advertising options to an end-user, and receives a reward any time the selection receives a click. This click-through-rate (CTR) paradigm of recommendation systems is standard. However, one challenge with obtaining good performance in this setting is that there are many more options than arm pulls. Therefore, there is incentive for one advertising firm to exchange information with another in order to gain a better understanding of CTR (Elena et al., 2021). In this case, one may pose the resulting ad targeting problem as a multi-agent MAB problem. Here, the existence of Byzantine agents may come by the presence of firms that seek to introduce offensive or divisive content into web services for non-economic incentives (Hinds et al., 2020). This issue has occurred several times in the past decade.

**Example 2 (Computer Networks)** In this setting, each agent in the network is a device or computer (Xiong & Jamieson, 2013; Ferdowsi et al., 2019), the arm selection specifies a non-local node in the network as a candidate which a data packet could be sent, meaning that the arm set is a subset of the node set. Reward is the outcome of a simple synchronize-acknowledge protocol (Syn-Ack), which is a binary indicator of whether a data packet was successfully received. The collaborative multi-agent aspect of this problem permits one to allow uploads within a Local Area Network (LAN), as in file-sharing protocols. It is natural to consider that a random subset of nodes could be Byzantine in this context, as malicious code or devices may attempt to deny the regular operation of a network. In this case, their presence would model a Distributed Denial of Service (DDOS) attack, or other similar cyber-attacks – see Gollmann (2010).

**Example 3 (Financial Markets)** Suppose a central treasury of a national government wants to stabilize its country's financial market. In this case, the arm pulled is an interest rate on treasury bonds which are typically held by large financial intuitions. The reward is the rate of increase/decrease in inflation, or another indicator of national economic health (such as labor force participation rate). In this context, there is an incentive to cooperate with the treasuries of other governments in order to achieve increased economic stability(Benigno & Benigno, 2002). The presence of Byzantine agents in this context manifests because not all governing bodies are interested in the economic stability of another. Indeed, in the presence of a geopolitical rival, decreasing economic performance may motivate communicating spurious or incorrect information to others (Macekura, 2020).

With the setting clarified, next we introduce our main algorithmic framework based upon UCB.

## 3   Byzantine-Resilient Collaborative UCB

To present our algorithm, we first need to introduce some notation. For each $i \in \mathcal{V}$ and $k \in \mathcal{M}$, we use $n_{i,k}(t)$ to denote the number of times agent $i$ has selected arm $k$ prior to time $t$. Moreover, denote $\bar{x}_{i,k}(t)$ as sample mean of arm $k$ reward tracked by $i$ at time $t$, i.e., $\bar{x}_{i,k}(t) = \frac{1}{n_{i,k}(t)} \sum_{\tau=0}^{t} \mathbb{1}(a_i(\tau) = k) X_{i,k}(\tau)$, where $\mathbb{1}(\cdot)$ is the indicator function which is 1 if the event in its argument is true and 0 otherwise.

We propose a protocol such that each agent $i \in \mathcal{V}$ transmits two scalars for each arm $k \in \mathcal{M}$ to all its out-neighbors: $n_{i,k}(t)$ and $\bar{x}_{i,k}(t)$. Due to the existence of Byzantine agents, we use $n_{ij,k}(t)$ and $\bar{x}_{ij,k}(t)$, $k \in \mathcal{M}$ to denote the possibly contaminated number of arm pulls and reward mean-estimates agent $i$ transmits to its out-neighbor $j$. If agent $i$ is normal, $n_{ij,k}(t) = n_{i,k}(t)$ and $\bar{x}_{ij,k}(t) = \bar{x}_{i,k}(t)$ for all $k \in \mathcal{M}$ and $j \in \mathcal{N}_i^-$. If agent $i$ is Byzantine, $n_{ij,k}(t)$ and $\bar{x}_{ij,k}(t)$ may be arbitrary for all $k \in \mathcal{M}$.

Before formalizing the procedure, we describe the key concepts behind its execution. The main idea is to use UCB as the decision-making policy. The key point of departure compared with single-agent UCB or non-Byzantine multi-agent UCB is that we have a filtering process to down-weight the effect of Byzantine agents. Recall that our goal is to design an algorithm that achieves full resilience and ensures no worse performance than the single-agent (non-cooperative) UCB. To this end, we construct a reward mean estimate $z_{i,k}(t)$ to be more accurate than the sample mean $\bar{x}_{i,k}(t)$ (the corresponding reward mean estimate in single-agent UCB). For resilient bandit, there are two factors that can affect the accuracy of $z_{i,k}(t)$: (i) the consistency of sample counts of agents in the neighborhood and (ii) the role of Byzantine agents.

**(i) Consistency Filter.** While UCB provides an effective solution for exploration in the single-agent case, the multi-agent case mandates that each agent not only explores each arm sufficiently itself but also requires certain local "consistency" conditions, i.e., the number of times an arm has previously been pulled is sufficient to construct a valid confidence set. To address this, a method to encourage a version of persistent exploration by forcing agents to select an insufficiently-explored arm if its corresponding sample count falls too much behind the network maximal sample count has been developed (Zhu & Liu, 2021). The upshot of this adjusted exploration scheme is that it can be shown to satisfy said consistency condition. However, it is inoperable in the Byzantine setting due to potentially spurious information. To alleviate this issue, we propose *threshold-consistency*, which instead of actively balancing the sample counts over the network, directly thresholds the number of neighbors an agent may employ to update the reward mean estimate by removing all the neighbors that have not explored sufficiently compared with itself. To be more precise, agents

may ensure local consistency in their confidence sets through a proper re-weighting of standard deviation in terms of $\mathcal{A}_{i,k}(t)$ neighbors that are considered post-winnowing. In particular, we introduce a truncation parameter $\kappa_i$, which parameterizes the "consistency level" (addressing issue (i)). If $\kappa_i$ is too large, then the consistency is insufficient, see equation 19; if $\kappa_i$ is set to be too small, then the restriction omits too many agents to obtain valid reward mean estimates. This tension may be formalized by restricting its viable range as $1 \leq \kappa_i < 2$. For effect on regret for different $\kappa_i \in [1, 2)$, see Figures 1 and 2.

**(ii) Trimmed Mean.** To achieve the nullification of Byzantine effect, we further propose a winnowing procedure on the reward mean-estimates in terms of an upper-bound on the number of Byzantine agents (addressing issue (ii)). That this is necessary may be observed by noting that even with the restricted construction of confidence sets, reward information from neighbors can be arbitrarily wrong. Therefore, no concentration bound may be employed to limit its impact. In this sense, to achieve resilience, normal agents need to negate the Byzantine effect, which may be achieved through the *trimmed-mean*. This procedure, similar to hard-thresholding, truncates the received sample means from other agents by omitting the $f$ largest and $f$ smallest values. This technique has found success in related resilient multi-agent consensus methods (Vaidya et al., 2012; Leblance et al., 2013; Saldaña et al., 2017; Saulnier et al., 2017). Through a re-parameterization (Lemma 2) and the threshold consistency previously mentioned, we can establish that the reward mean estimate is actually superior to its single-agent counterpart. It is worth emphasizing that the trimmed-mean idea has to be incorporated with a suitably chosen consensus variable for resilient reward mean estimation; see Remark 2 and Example 4 in the Appendix.

With these elements properly motivated, we are ready to present the main algorithm for decentralized UCB with Byzantine agents.

---

**Algorithm 1: Filter**$(i, k, t)$: Consistency and trimmed mean filters of agent $i$ on arm $k$ at time $t$

**Input:** agent $i$, $\kappa_i$, arm $k$, time $t$

**1** Set $\tilde{\mathcal{A}}_{i,k}(t) = \mathcal{N}_i$

**2 for** $j \in \mathcal{N}_i$ **do**

**3**      **if** $\kappa_i n_{ij,k}(t) < n_{i,k}(t)$ **then**

**4**          Remove $j$ from $\tilde{\mathcal{A}}_{i,k}(t)$                  // Consistency-Filter

**5**      **end**

**6 end**

**7** Set $\mathcal{A}_{i,k}(t) = \tilde{\mathcal{A}}_{i,k}(t)$

**8 if** $|\mathcal{A}_{i,k}(t)| \leq 2f$ **then**

**9**      $z_{i,k}(t) = \bar{x}_{i,k}(t)$

**10 else**

**11**      Set $\tilde{\mathcal{B}}_{i,k}(t) = \mathcal{A}_{i,k}(t)$

**12**      Sort $\bar{x}_{ji,k}(t)$ in descending order for $j \in \tilde{\mathcal{B}}_{i,k}(t)$ and removes the indices corresponding to the

**13**      $f$ largest and $f$ smallest values from $\tilde{\mathcal{B}}_{i,k}(t)$              // Trimmed-Mean

**14**      Set $\mathcal{B}_{i,k}(t) = \tilde{\mathcal{B}}_{i,k}(t)$

**15**      $z_{i,k}(t) = \frac{1}{|\mathcal{B}_{i,k}(t)|+1}\left(\bar{x}_{i,k}(t) + \sum_{j \in \mathcal{B}_{i,k}(t)} \bar{x}_{ji,k}(t)\right)$

**16 end**

---

**Initialization:** At initial time $t = 0$, each normal agent $i \in \mathcal{H}$ samples each arm $k$ exactly once and then sets $n_{i,k}(0) = 1$ and $\bar{x}_{i,k}(0) = X_{i,k}(0)$.

Between clock times $t$ and $t + 1$, with $t \in \{0, 1, \ldots, T\}$, each normal agent $i \in \mathcal{H}$ performs the steps enumerated below in the order indicated.

**Transmitting:** Agent $i$ transmits the possibly contaminated number of arm pulls $n_{ij,k}(t)$ and reward mean-estimate $\bar{x}_{ij,k}(t)$, $k \in \mathcal{M}$ to each of its out-neighbors $j \in \mathcal{N}_i^-$ and meanwhile receives $n_{hi,k}(t)$ and $\bar{x}_{hi,k}(t)$, $k \in \mathcal{M}$ from each of its neighbors $h \in \mathcal{N}_i$.

**Filtering:** Agent $i$ performs two steps to possibly filter out the malefactors in its neighbor set $\mathcal{N}_i$.

Step A [Consistency Filter]: For each arm $k \in \mathcal{M}$, agent $i$ filters out those neighbors' indices $j$ from $\mathcal{N}_i$ for which $\kappa_i n_{ji,k}(t) < n_{i,k}(t)$, where $\kappa_i \in [1,2)$ is a constant, and sets the remaining index set as $\mathcal{A}_{i,k}(t)$, i.e.,

$$\mathcal{A}_{i,k}(t) = \big\{ j \in \mathcal{N}_i : \kappa_i n_{ji,k}(t) \geq n_{i,k}(t) \big\}.$$

Step B [Trimmed Mean Filter]: For each arm $k \in \mathcal{M}$, if $|\mathcal{A}_{i,k}(t)| > 2f$, agent $i$ filters out those neighbors' indices $h$ from $\mathcal{A}_{i,k}(t)$ whose $\bar{x}_{hi,k}(t)$ are the $f$ largest and $f$ smallest among $\bar{x}_{ji,k}(t)$, $j \in \mathcal{A}_{i,k}(t)$, with ties broken arbitrarily, and then sets the remaining index set as $\mathcal{B}_{i,k}(t)$, otherwise agent $i$ sets $\mathcal{B}_{i,k}(t) = \emptyset$. To be more precise, let $\pi$ be any non-decreasing permutation of $\mathcal{A}_{i,k}(t)$ for which $\bar{x}_{\pi(h)i,k}(t) \leq \bar{x}_{\pi(h+1)i,k}(t)$ for all $h \in \{1, \ldots, |\mathcal{A}_{i,k}(t)| - 1\}$. Then,

$$\mathcal{B}_{i,k}(t) = \begin{cases} \emptyset, & \text{if } |\mathcal{A}_{i,k}(t)| \leq 2f, \\ \big\{ \bar{x}_{\pi(h)i,k}(t) : f+1 \leq h \leq |\mathcal{A}_{i,k}(t)| - f \big\}, & \text{else.} \end{cases} \tag{2}$$

**Decision Making:** Agent $i$ calculates its current estimate of reward mean $\mu_k$ for each arm $k \in \mathcal{M}$ as

$$z_{i,k}(t) = \frac{1}{|\mathcal{B}_{i,k}(t)| + 1} \left( \bar{x}_{i,k}(t) + \sum_{j \in \mathcal{B}_{i,k}(t)} \bar{x}_{ji,k}(t) \right), \tag{3}$$

and based upon this information, computes its exploration bonus (derived from Hoeffding in Lemma 3 in the Appendix) from each arm $k$ as

$$C\big(t, n_{i,k}(t)\big) = \sqrt{\frac{2 g_{i,k}(t) \log t}{n_{i,k}(t)}},$$

where the variance term is defined in terms of the proportion of arm pulls remaining post-filtering $|B_{i,k}(t)|$ as

$$g_{i,k}(t) = \begin{cases} 1, & \text{if } \mathcal{B}_{i,k}(t) = \emptyset, \\ \dfrac{\kappa_i}{4} + \dfrac{\kappa_i}{4(|\mathcal{B}_{i,k}(t)| + 1)} + \dfrac{1}{(|\mathcal{B}_{i,k}(t)| + 1)^2}, & \text{else.} \end{cases} \tag{4}$$

Then, the arm $a_i(t+1)$ that maximizes the Byzantine-filtered upper-confidence bound for agent $i$ at time $t+1$ is selected

$$a_i(t+1) = \arg\max_{k \in \mathcal{M}} \big( z_{i,k}(t) + C(t, n_{i,k}(t)) \big).$$

**Updating:** Agent $i$ updates its variables as

$$n_{i,k}(t+1) = \begin{cases} n_{i,k}(t) + 1 & \text{if } k = a_i(t+1), \\ n_{i,k}(t) & \text{if } k \neq a_i(t+1), \end{cases}$$

$$\bar{x}_{i,k}(t+1) = \frac{1}{n_{i,k}(t+1)} \sum_{\tau=0}^{t+1} \mathbb{1}(a_i(\tau) = k) X_{i,k}(\tau).$$

These steps are summarized as Algorithm 2 in the Appendix.

**Remark 1 (Communication Cost)** Each agent to broadcast two variables: one real number and one integer for each arm at each time step. The communication cost is comparable to many existing decentralized cooperative MAB algorithms (Landgren et al., 2016; Landgrena et al., 2021; Zhu & Liu, 2021). A simple way to reduce the communication cost of Algorithm 1 is to allow agents to pick a subset of neighbors with which to communicate. Observe from Theorem 3 that no network connectivity requirement is needed to ensure performance comparable to the single-agent counterpart. Using communication epochs with a fixed constant length is an alternative approach (Martínez-Rubio et al., 2019; Dubey & Pentland, 2020)), where each agent only communicates and makes arm decisions at the start of the epoch, and keeps selecting the arm until the end of the phase. Doing so yields a larger constant in the regret analysis of Theorem 2.

**Remark 2 (On the insufficiency of consensus and its variants)** Consider related algorithms under "full arm observability" setting in Landgren et al. (2016); Landgrena et al. (2021); Martínez-Rubio et al. (2019); Zhu & Liu (2021): each agent explores the entire arm set and then decides which arm to select. In this setting, agents arm selection strategies exhibit incompatibility with Byzantine filtering such as the trimmed-mean method. These methods build upon the consensus protocol, which incorporates the weighted average of neighbors' reward mean estimates in the previous step with reward information at current step in the updating of the reward mean estimate. For the non-Byzantine setting, such a technique propagates information over the network and ensures the reward mean estimate tends towards consistency as time progresses. In the presence of Byzantine agents, if one employs a running consensus on $z_{i,k}(t)$, it forms one component of $z_{i,k}(t+1)$, which causes each step to accumulate the bias from the previous (which can be shown in equation 9 in Lemma 1). We note that the trimmed mean does not directly cancel out the Byzantine effect, that is, $z_{i,k}(t)$ can still be biased. Algorithm 1 achieves resilience in that when $z_{i,k}(t+1)$ is updated, $z_{i,k}(t)$ is not used. Then, the bias of $z_{i,k}(t)$ due to Byzantine agents does not accumulate across time. Thus, as normal agents have increasingly more accurate sample means as they accumulate samples, the potential effect of Byzantine attacks shrinks, eventuating in the bias converging to null. Detailed discussion can be found in Appendix E.

## 4 Sublinear Regret in Presence of Byzantine Agents

This section presents our key theoretical results, which establish the sublinear regret of the algorithm presented above. We begin with lower and upper bounds of each normal agent's regret.

**Theorem 1 (Lower Bound)** *The expected cumulative regret of any normal agent $i \in \mathcal{H}$ satisfies*

$$\liminf_{T \to \infty} \frac{R_i(T)}{\log T} \geq O\Big(\frac{1}{\max\{|\mathcal{N}_i| - 2f + 1, 1\}}\Big).$$

Observe from the above theorem that when a normal agent has at least $2f + 1$ neighbors, the lower bound on its regret is actually better than the single-agent UCB1 algorithm (Auer et al., 2002a). We clarify the form of this gap next. To do so, denote as $\Delta_k \overset{\Delta}{=} \mu_1 - \mu_k$ the gap between the largest mean and the mean for each arm $k \in \mathcal{M}$.

**Theorem 2 (Upper Bound)** *The expected cumulative regret of any normal agent $i \in \mathcal{H}$ satisfies*

$$R_i(T) \leq \min_{\tau \in \{1,...,T\}} \left( \sum_{k: \Delta_k > 0} \left( \max_{t \in \{1,...,\tau\}} \frac{8g_{i,k}(t) \log t}{\Delta_k} + \Big(1 + \frac{\pi^2}{3}\Big)\Delta_k \right) + (T - \tau)\Delta_M \right)$$

$$\leq \sum_{k: \Delta_k > 0} \left( \max_{t \in \{1,...,T\}} \frac{8g_{i,k}(t) \log t}{\Delta_k} + \Big(1 + \frac{\pi^2}{3}\Big)\Delta_k \right). \tag{5}$$

Note that the regret upper bound in Theorem 2 depends on $g_{i,k}(t)$ whose definition equation 4 is influenced by $\mathcal{B}_{i,k}(t)$ and $\kappa_i$. The construction of set $\mathcal{B}_{i,k}(t)$ in equation 2 implies that, in general, a larger $f$ and a smaller $|\mathcal{N}_i|$ correspond to a larger regret bound. The influence of $\kappa_i$ on $g_{i,k}(t)$ is not monotone; see Figures 1–2 and its discussion in **(i) Consistency Filter** in the preceding section. The following results are the performance comparisons with the single-agent (non-cooperative) UCB1 (Auer et al., 2002a, Theorem 1), where the upper bound of each agent's regret was establised as $\sum_{k: \Delta_k > 0} \big(\frac{8 \log T}{\Delta_k} + (1 + \frac{\pi^2}{3})\Delta_k\big)$.

**Theorem 3 (Per-agent Outperformance)** *The regret upper bound for each normal agent $i \in \mathcal{H}$ is always no worse than that of the single-agent UCB1, i.e.,*

$$R_i(T) \leq \sum_{k: \Delta_k > 0} \left( \frac{8 \log T}{\Delta_k} + \Big(1 + \frac{\pi^2}{3}\Big)\Delta_k \right).$$

Note that Theorems 1–3 do not rely on any graphical conditions like connectivity and local degrees, which is a departure from prior results (Vial et al., 2021; 2022; Mitra et al., 2022). The following theorem shows that if certain local degree condition is satisfied, all the normal agents can collectively outperform the non-cooperative case, which is still independent of global connectivity.

**Theorem 4 (Network Outperformance)** *If each agent has at least $3f+1$ neighbors, then the cumulative regret upper bound of all normal agents is strictly better than that of the non-cooperative counterpart:*

$$R(T) = \sum_{i \in \mathcal{H}} R_i(T) < |\mathcal{H}| \sum_{k:\, \Delta_k > 0} \left( \frac{8 \log T}{\Delta_k} + \left( 1 + \frac{\pi^2}{3} \right) \Delta_k \right). \tag{6}$$

The comparison of the above results with the existing literature can be found in Appendix C. The proofs of the above theorems are given in Appendix F. Here we provide a sketch of the proofs.

**Sketch of Proofs:** To show the two outperformance results in Theorem 3 and Theorem 4, we first detail the regret upper bound in Theorem 2. To begin with, we estimate the value of $\mathbf{E}(n_{i,k}(T))$ as each agent's regret satisfies $R_i(T) = \sum_{k:\Delta_k>0} \mathbf{E}(n_{i,k}(T))\Delta_k$. Towards this end, we first follow the logic flow of the standard single-agent UCB1 algorithm (Auer et al., 2002a, Proof of Theorem 1), which makes use of the decision-making step in equation 11 and turns the problem into estimating the concentration bounds of the reward mean estimate equation 13. That is where we depart from the single-agent analysis, as we have a more complicated design of the reward mean estimate. We proceed by dividing the analysis into two cases based on the number of neighbors retained after the filtering steps to estimate the concentration bound, which utilizes the Hoeffding's inequality (Lemma 3). To make Hoeffding's inequality apply, there are two important steps: we slice the random sample count to all possible values (see e.g., equation 14), the other is using Lemma 2 to *restrict the Byzantine behavior* with the help of Trimmed Mean Filter (Step B). In addition, to obtain a tight concentration bound, we make use of the *local exploration consistency* assured by Consistency Filter (Step A) in the computation of Hoeffding's inequality, see equation 16. In this way, our analysis contains fundamental steps that are not present in prior analyses of UCB, and addresses fundamental challenges associated with the Byzantine effect. Therefore, we can obtain equation 5, which implies Theorem 3, by showing the adjusted variance [cf. equation 4] $g_{i,k}(t) \leq 1$ for all $i \in \mathcal{H}, k \in \mathcal{M}$ and $t = 1, \ldots, T$. Then, it suffices to show there exists at least one agent have strictly better performance on each arm at each time for Theorem 4.

## 4.1 Time-varying Random Graphs

This subsection extends theoretical results to the cases when neighbor graph $\mathbb{G}(t)$ changes over time. Define $\mathcal{N}_i(t)$ as $\mathcal{N}_i^-(t)$ as the corresponding neighbor set and out-neighbor set of agent $i$ at time $t$. Replacing $\mathcal{N}_i$ and $\mathcal{N}_i^-$ by $\mathcal{N}_i(t)$ and $\mathcal{N}_i^-(t)$, respectively, in Algorithm 2, yields a Byzantine-resilient decentralized bandit algorithm for time-varying graphs. Specifically, the results of per-agent regret stated in Theorems 1–3 still hold with exactly the same analyses. This is because when we estimate the reward mean estimate $z_{i,k}(t)$, we only make use of $\bar{x}_{j,k}(t)$ where $j \in \mathcal{N}_i(t)$, which is one-time local information, and thus its consequences do not rely on graph topology variation over time. Formal definitions regarding time-varying graphs are provided in Appendix D. The following theorem shows that Algorithm 2 possesses collective outperformance over the non-cooperative case for time-varying random graphs, provided a probabilistic local degree condition is satisfied.

**Theorem 5 (Network Outperformance)** *If the probability that every agent in the network has at least $3f+1$ neighbors is $p \in (0, 1]$ at each time $t$, then the network total regret upper bound is strictly better than the non-cooperative counterpart, i.e., equation 6 holds.*

We note that this is the first sublinear regret result for MAB in the decentralized setting in the presence of Byzantine ageants, to the best of our knowledge.

**Remark 3** From the proof of Theorem 4 in Appendix F, the outperformance over the single-agent counterpart over fixed graphs when the $3f+1$ degree requirement is logarithmic, i.e., we have a smaller coefficient

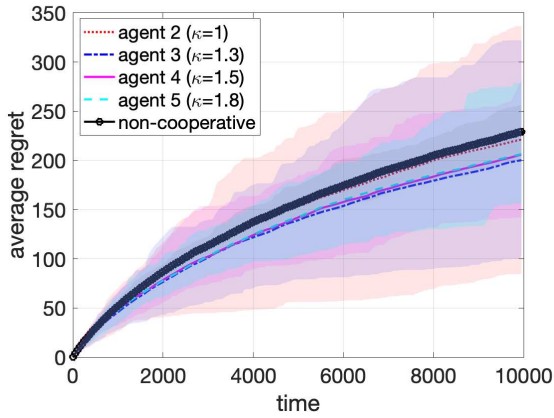

Figure 1: Per-agent regret for each normal agent with heterogeneous $\kappa_i$

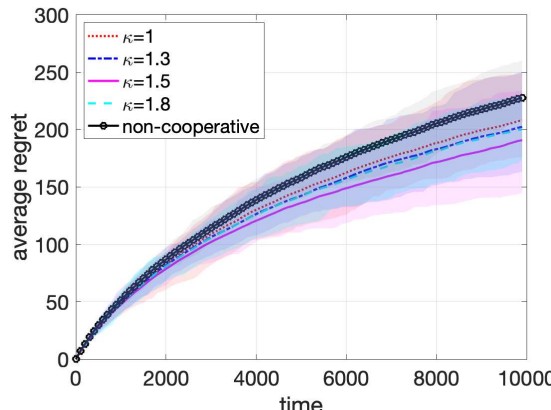

Figure 2: Network averaged regret for normal agents with homogeneous $\kappa$

of the $\log T$ term. The same holds for Theorem 5 with $p = 1$ because the analysis for Theorem 4 can be directly applied to this case. Yet for random graphs with $p \in (0, 1)$, the outperformance is of order $O(1)$ from the proof of Theorem 5.

## 5 Numerical Evaluation

We conduct experiments – additional studies are in Appendix G. We consider a four-arm bandit problem whose arm distributions are Bernoulli with mean $0.5, 0.45, 0.4, 0.3$, respectively. The Byzantine agent broadcasts $0.4, 0.5, 0.4, 0.3$ as the corresponding reward information of the four arms to all the normal agents, and sample count $n_{1,k}(t) = n_{i,k}(t)$ to all normal agents $j$ with $i$ being randomly selected in $\{2, 3, 4, 5\}$. It can be shown in Appendix E that the same or similar Byzantine policy is sufficient to make the existing decentralized bandit algorithms not suitable to find an optimal arm. The graph in our experiments follows a random structure where the probability each directed edge is activated is a common value $q$. The total time $T$ is set as $T = 10000$. Figure present sample means and standard shaded deviations over 50 runs.

We first conduct two simulations to illustrate the effect of threshold-consistency parameter $\kappa_i$ under a five-agent network with only agent 1 being Byzantine over a random graph generated with $q = 0.8$. Figure 1 shows the comparison of per-agent regret of each normal agent $i$ with a different $\kappa_i$ and the non-cooperative counterpart (Auer et al., 2002a) and Figure 2 presents the network total performance of all normal agents when they use a same $\kappa$ together with the non-cooperative counterpart (Auer et al., 2002a). Observe that our algorithm outperforms the non-cooperative counterpart in terms of both the network regret and the individual regret of each normal agent, which corroborates Theorems 3 and 4. Moreover, it appears that the $3f + 1$ degree requirement (which requires the graph to be complete for a five-agent network) is not necessary for superior performance in practice. In addition, the regret does not appear to be monotone in terms of $\kappa_i$, which is consistent with the discussion of $\kappa_i$ in Section 3.

We next test the network performance over a ten-agent network with two Byzantine agents to study the effect of $q$, which is related to the expected degree of the network. The simulation result over a fixed graph is shown in Figure 3. For time-varying graphs, note that when $q = 1$, the graphs are (fixed) complete graphs, and when $0 < q < 1$, there is a relationship between $q$ and $p$, the probability of the degree requirement is met, which is defined in Theorem 5, $p = [\sum_{i=7}^{9} \binom{9}{i} q^i (1-q)^{9-i}]^{10}$. The simulation result is shown in Figure 4. Observe that larger $q$ results in better performance. Moreover, the normal agents' performance using Algorithm 1 is no worse than that of the single-agent counterpart, even with a small $q$, which substantiates Theorem 3. However, although for time-varying random graphs, Theorem 5 holds a positive result in terms of comparison with the non-cooperative counterpart for all $0 < p \le 1$, experimentally we see that in some

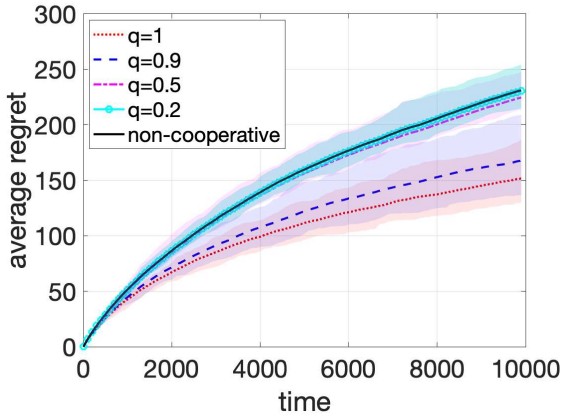

Figure 3: Network averaged regret for normal agents under fixed random graphs with various values of $q$

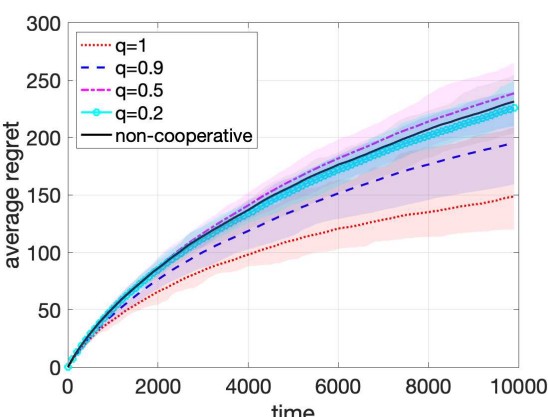

Figure 4: Network averaged regret for normal agents under time-varying random graphs with various values of $q$

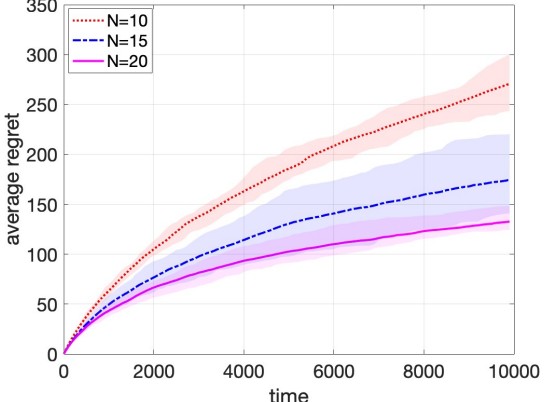

Figure 5: Network averaged regret for normal agents with various values of $N$

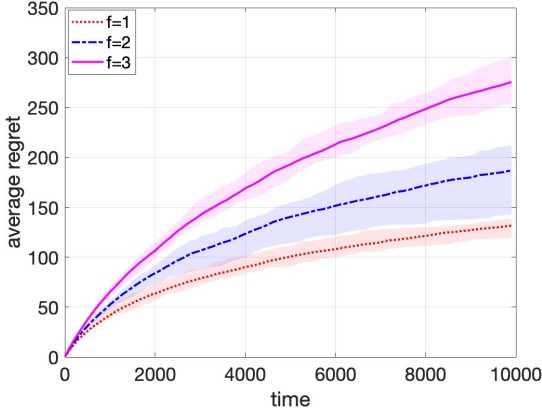

Figure 6: Network averaged regret for normal agents with various values of $f$

cases our algorithm returns a worse performance in experiments, illuminating a possibly theory-practice gap. Furthermore, unlike the fixed graph setting, the regret appears to be not decreasing in terms of $q$.

We then compare the algorithm performance of multi-agent networks with distinct network size $N$ or the number of Byzantine agents $f$. Consider neighbor graphs, which are random graphs with a common $q = 0.8$, and set all $\kappa_i$ to a common value of 1.3. We first fix $f = 2$ and test the performance of the agents for $N = 10, 15, 20$ respectively; the simulation results are shown in Figure 5. Then, we fix $N = 15$ and test the performance of the agents for $f = 1, 2, 3$ respectively, as illustrated in Figure 6. It can be seen that the regret increases with $f$ and decreases with $N$. Since, in a random graph scenario, the average/expected number of an agent's neighbors increases with $N$, these observations are consistent with the theoretical results discussed immediately after Theorem 2.

Finally, we compare the actual regret with the theoretical upper bound. We set $\kappa_i$ to a uniform value of 1.3, $N = 10$, $f = 2$, and $q = 0.8$; the simulation results are presented in Figure 7. Although there appears to be a nontrivial gap between the regret upper bound and the actual regret observed with our algorithm, this is not surprising. Our algorithm is a decentralized multi-agent generalization of the single-agent UCB1 algorithm, which exhibits similar performance. The gap likely arises because the theoretical analysis considers the

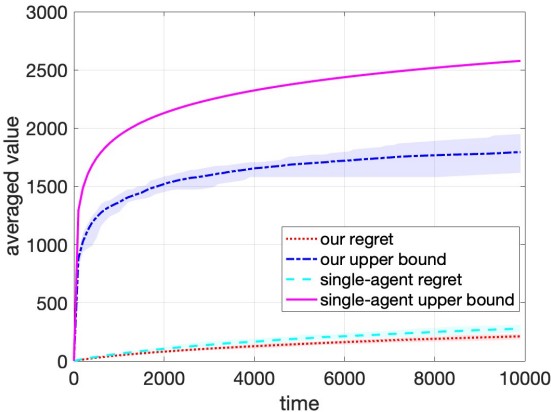

Figure 7: Network averaged regret v.s. network averaged regret upper bound

"worst-case" performance, which rarely occurs in practice and is difficult to identify and evaluate due to the complex statistical processes involved.

## 6    Conclusion

We considered cooperative multi-armed bandit problems in the presence of Byzantine agents. To solve this problem, we developed a consensus-type information mixing strategy that operates in tandem with a filtering scheme that contains two stages: limit the number of possible neighbors with which to exchange information, and threshold extreme values of the reward mean estimate. The result achieves regret that is certifiably better than an individual agent. Numerical experiments demonstrate the merits of this approach in practice. Limitations of this framework include: (1) it is not yet capable of incorporating contextual information into arm selection, (2) the obtained regret upper bound is not monotone with respect to $\kappa_i$, requiring extra effort to tune this parameter, and (3) there is a potential gap between theoretical and practical results in the time-varying random graphs scenario.

### Acknowledgments

J. Zhu is currently with Zhejiang Lab and was previously affiliated with the Department of Applied Mathematics and Statistics at Stony Brook University. The majority of the work was completed while J. Zhu was at Stony Brook University. The work of J. Liu was supported by the National Science Foundation under Grant No. 2230101.

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

## A  Pseudocode

---

**Algorithm 2:** Resilient Decentralized UCB

---

**Input:** $\mathbb{G}, T, \kappa_i$

**1 Initialization** Each agent samples each arm exactly once. Initialize $z_{i,k}(0) = \bar{x}_{i,k}(0) = X_{i,k}(0)$, $m_{i,k}(0) = n_{i,k}(0) = 1$

**2 for** $t = 0, \ldots, T$ **do**

**3**  $\quad$ **for** $k = 1, \ldots, M$ **do**

**4**  $\quad\quad$ Agent $i$ runs **Filter**$(i, k, t)$

**5**  $\quad$ **end**

**6**  $\quad a_i(t+1) = \arg\max_{k \in \mathcal{M}} \left( z_{i,k}(t) + C(t, n_{i,k}(t)) \right)$  $\qquad$ // decision making

**7**  $\quad n_{i,k}(t+1) = n_{i,k}(t), \ \forall k \in [M]$

**8**  $\quad n_{i,a_i(t+1)}(t+1) = n_{i,a_i(t+1)}(t) + 1$  $\qquad$ // information updating

**9**  $\quad \bar{x}_{i,k}(t+1) = \frac{1}{n_{i,k}(t+1)} \sum_{\tau=0}^{t+1} \mathbb{1}(a_i(\tau) = k) X_{i,k}(\tau)$

**10**  $\quad$ Agent $i$ sends $n_{ij,k}(t+1) = n_{i,k}(t)$ and $\bar{x}_{ij,k}(t) = \bar{x}_{i,k}(t), \forall k \in [M]$ to $j \in \mathcal{N}_i^-$

**11**  $\quad$ Agent $i$ receives $n_{ji,k}(t), \bar{x}_{ji,k}(t), \forall k \in [M]$ from $j \in \mathcal{N}_i$  $\qquad$ // information propagation

**12 end**

---

## B  Related Work

Resilience in multi-agent optimization dates at least back to Su & Vaidya (2016). Many prior works in this area formalize tradeoffs between resilience and accuracy (Kuwaranancharoen et al., 2020; Yang & Bajwa,

2019; Fang et al., 2022; He et al., 2022; Wu et al., 2023; Su & Vaidya, 2016; Sundaram & Gharesifard, 2018; Su & Vaidya, 2020; Zhao et al., 2020), i.e., the limit point exhibits bias with respect to the optimizer dependent on the extent of adversarial manipulation. To refine this dependence, assumptions that there is "objective function redundancy" have been considered (Gupta et al., 2021; Gupta & Vaidya, 2021; Kaheni et al., 2022) for the special case of a complete graph. Extensions to tree graphs (federated setting) have also been studied (Liu et al., 2021b; Gupta et al., 2023). Significant attention has been paid to designing Byzantine-resilient variants of stochastic gradient iteration (Chen et al., 2017; Blanchard et al., 2017; Alistarh et al., 2018; Su & Xu, 2019; Chen et al., 2018) – see Yang et al. (2020) for a review.

Designing resilient bandit algorithms may be traced back to a single-agent setting in Auer et al. (2002b), where an EXP3 with $O(\sqrt{T})$ regret was established. Follow-on work (Bubeck & Cesa-Bianchi, 2012; Audibert & Bubeck, 2009; Auer & Chiang, 2016) refined the regret constants. Subsequent work (Jun et al., 2018; Liu & Shroff, 2019) established that with $O(\log T)$ attack cost, UCB and greedy algorithms can suffer linear regret $O(T)$. Guan et al. (2020) established that a $O(\log T)$ regret can be achieved under certain assumption on reward distributions with a median-based technique. Subsequent extensions of adversarial models to contextual bandits have been studied Slivkins (2011); Wang et al. (2022b); Kwon et al. (2022). In recent years, Byzantine robust reinforcement learning for federated setting is also developed Chen et al. (2023); Jadbabaie et al. (2022); Fan et al. (2021).

Most similar to the setting studied here are models of adversarial manipulation of MAB methods in the multi-agent setting. Dubey & Pentland (2022) discussed a Byzantine setting where at each time agents observe a true reward with probability $1 - \epsilon$ and observes a reward from an unknown but fixed distribution with probability $\epsilon$. Madhushani et al. (2021); Wang et al. (2022a); Liu et al. (2021a) considered an adaptive Byzantine corruption where any reward when transmitted can be corrupted the adversary. The setting we study is similar to Vial et al. (2021; 2022); Mitra et al. (2022), Mitra et al. (2022) is specialized for linear bandit problems and it discussed a solution methodology that requires centralization, whereas Vial et al. (2021; 2022) partition the set of arms into distinct subsets that are assigned among the agents. To achieve robustness, a connected and undirected normal network is required, in contrast to our setting. These methods mandate that the proportion of Byzantine agents relative to normal agents is sufficiently small, in order to outperform a single-agent bandit method. In particular, (Vial et al., 2022, Theorem 2) contains the possibility that performance can be worse than the single-agent counterpart, and always exhibits degradation in regret bound under our arm aligning setting which is widely considered in multi-armed bandits literature (e.g. Landgren et al. (2016); Landgrena et al. (2021); Martínez-Rubio et al. (2019); Zhu et al. (2021); Zhu & Liu (2021)). A more rigorous contrast of the regret results in these works may be found in the succeeding section.

## C    Comparison With Existing Literature

### C.1    Comparison with decentralized stochastic bandit algorithms

In the context of decentralized algorithms within a stochastic bandit setting, as described in prior works such as Landgren et al. (2016); Landgrena et al. (2021); Martínez-Rubio et al. (2019); Zhu et al. (2021); Zhu & Liu (2021), it is observed that most of these algorithms impose specific network connectivity requirements even in the absence of Byzantine attacks. Furthermore, the majority of these algorithms do not guarantee superior individual performance compared to their non-cooperative counterparts, with the exception of Zhu & Liu (2021). In contrast, our proposed algorithm eliminates the need for any network connectivity prerequisites while ensuring individual performance that is at least as good as that of the non-cooperative counterpart.

### C.2    Comparison with resilient algorithms

In this subsection, we give a detailed comparison with the existing resilient bandit algorithms Mitra et al. (2022); Vial et al. (2021; 2022), which are under the similar Byzantine setting with ours.

In the work of Mitra et al. (2022), the authors investigate a robust linear bandit problem within the context of federated learning. This scenario necessitates the presence of a central server capable of managing all

the communications. The study presents a regret bound of $O(\sqrt{T})$, and this regret can be further improved when the proportion of Byzantine agents decreases or when the overall number of agents increases. In contrast, our algorithm is designed for a decentralized communication scenario, which is more versatile in terms of the types of communication graphs it can accommodate. It is worth noting that considering the federated setting effectively implies an assumption that the underlying communication graph takes on either a complete graph or a star graph configuration, with the central node (or root) being consistently reliable. These configurations are considered special because they entail the presence of at least one normal agent capable of establishing communication with all other agents. Consequently, the methodology proposed by Mitra et al. (2022) finds limited applicability in our context. Nevertheless, like Mitra et al. (2022), our algorithm also affords agents the opportunity to enhance their performance in response to specific changes in the network structure. As detailed in Equation equation 5, each normal agent generally experiences a lower regret when it possesses a larger number of neighbors or when $f$ decreases. This property is also explained in the subsequent paragraph following Theorem 2. Furthermore, in the overlapping scenario analyzed in both Mitra et al. (2022) and our paper, where agents engage in federated/centralized communication to address a stochastic bandit problem, our algorithm achieves a superior asymptotic regret bound. Specifically, the regret bound in Mitra et al. (2022) is at least $O(\sqrt{T})$, which is higher than our regret bound of $O(\log T)$.

In the work of Vial et al. (2021; 2022), the authors explore a scenario where each agent is exclusively responsible for exploring a subset of arms. While this setting may initially appear to encompass a more generalized arm alignment setup compared to ours, it imposes a requirement for a connected and undirected network among normal agents within the communication graph. This particular requirement diminishes its practicality from a design perspective. Furthermore, if we examine the specific case where both papers allow each agent to explore the entire set of arms, it becomes evident that both Vial et al. (2021) and Vial et al. (2022) present regret bounds that are inferior to those of the non-cooperative counterparts. To elaborate, both (Vial et al., 2021, Theorem 2) and (Vial et al., 2022, Theorem 2) reveal that the coefficient associated with the $\log T$ term in their respective regret upper bounds is given by $\sum_{k:\Delta_k>0} \frac{4\alpha}{\Delta_k}$ with a parameter $\alpha$ strictly greater than 2. In contrast, the single-agent UCB algorithm achieves a coefficient of $\sum_{k:\Delta_k>0} \frac{8}{\Delta_k}$ for the same term. Consequently, it becomes evident that the algorithms proposed by Vial et al. (2021; 2022) fail to surpass the performance of the classic single-agent counterpart within the studied setting.

## D  Time-varying Graphs

**Theorem 6 (Lower Bound)** *For any time-varying neighbor graph sequence $\{\mathbb{G}(t)\}$, the expected cumulative regret of any normal agent $i \in \mathcal{H}$ satisfies*

$$\liminf_{T\to\infty} \frac{R_i(T)}{\log T} \geq O\Big(\frac{1}{\max\{\max_{t\in\{1,\dots,T\}} |\mathcal{N}_i(t)| - 2f + 1, 1\}}\Big).$$

**Theorem 7 (Upper Bound)** *For any time-varying neighbor graph sequence $\{\mathbb{G}(t)\}$, the expected cumulative regret of any normal agent $i \in \mathcal{H}$ satisfies*

$$R_i(T) \leq \min_{\tau\in\{1,\dots,T\}} \left( \sum_{k:\,\Delta_k>0} \left( \max_{t\in\{1,\dots,\tau\}} \frac{8g_{i,k}(t)\log t}{\Delta_k} + \Big(1 + \frac{\pi^2}{3}\Big)\Delta_k \right) + (T-\tau)\Delta_M \right)$$

$$\leq \sum_{k:\,\Delta_k>0} \left( \max_{t\in\{1,\dots,T\}} \frac{8g_{i,k}(t)\log t}{\Delta_k} + \Big(1 + \frac{\pi^2}{3}\Big)\Delta_k \right).$$

**Theorem 8 (Per-agent Outperformance)** *For any time-varying neighbor graph sequence $\{\mathbb{G}(t)\}$, the regret upper bound for each normal agent $i \in \mathcal{H}$ is always no worse than that of the single-agent UCB1, i.e.,*

$$R_i(T) \leq \sum_{k:\,\Delta_k>0} \left( \frac{8\log T}{\Delta_k} + \Big(1 + \frac{\pi^2}{3}\Big)\Delta_k \right).$$

The above three theorems can be proved using the same argument as in the proofs of Theorems 1–3, respectively.

# E   Counterexample

In this appendix, we will take the updating policy in Zhu & Liu (2021) as an example to show that using running consensus, a commonly used updating policy in the literature, cannot nullify the Byzantine effect even with a filtering process, rather, it makes the reward mean estimate inaccurate.

We begin with introducing the algorithm in Zhu & Liu (2021). Denote the reward mean estimate as $\tilde{z}_{i,k}(t)$, which is updated as follows:

$$\tilde{z}_{i,k}(t+1) = \frac{1}{|\mathcal{N}_i|} \sum_{j \in \mathcal{N}_i} \tilde{z}_{j,k}(t) + \bar{x}_{i,k}(t+1) - \bar{x}_{i,k}(t), \tag{7}$$

where $\bar{x}_{i,k}(t)$ denotes the sample mean, i.e.,

$$\bar{x}_{i,k}(t) = \frac{1}{n_{i,k}(t)} \sum_{\tau=0}^{t} \mathbb{1}(a_i(\tau) = k) X_{i,k}(\tau).$$

Initial $\tilde{z}_{i,k}(0)$ is set equal to $X_{i,k}(0)$. We omit the decision-making process studied in Zhu & Liu (2021) here. This is because, as we will show in later discussions, under the resilient setting, using such an updating policy at agent $i$ cannot even ensure an accurate $\tilde{z}_{i,k}(t)$. Thus, regardless of the decision-making policy, the algorithm results in linearly escalating regret.

Under the resilient setting studied in this paper, for a fair comparison, we add a trimmed-mean filtering process before equation 7 to enable the algorithm to handle certain Byzantine effects. Let $\tilde{z}_{ij,k}(t) = \tilde{z}_{i,k}(t)$ if $i \in \mathcal{H}$ and be an arbitrary value if $i \in \mathcal{F}$. At each time step, each agent $i \in \mathcal{H}$ transmits $\tilde{z}_{ij,k}(t)$ to each of its out-neighbors $j \in \mathcal{N}_i^-$ and simultaneously receives $\tilde{z}_{hi,k}(t)$ from each of its neighbors $h \in \mathcal{N}_i$; the agent then sorts all received $\tilde{z}_{hi,k}(t)$, $h \in \mathcal{N}_i$ in descending order and filters out the largest $f$ and the smallest $f$ values. Let $\mathcal{M}_{i,k}(t)$ be the retained neighbor set of agent $i$ after filtering at time $t$. Then, agent $i$ updates its reward mean estimate $\tilde{z}_{i,k}(t)$ as

$$\tilde{z}_{i,k}(t+1) = \frac{1}{|\mathcal{M}_{i,k}(t)|} \sum_{j \in \mathcal{M}_{i,k}(t)} \tilde{z}_{ji,k}(t) + \bar{x}_{i,k}(t+1) - \bar{x}_{i,k}(t). \tag{8}$$

Here equation 8 serves as the modified updating policy in Zhu & Liu (2021) under the resilient bandit setting. We will show in the following example that it results in $|\mathbf{E}(\tilde{z}_{i,k}(t)) - \mu_k|$ being always bounded below by a strictly positive constant for any $n_{i,k}(t) \in \{1, \ldots, t\}$. This fact implies that the reward mean estimate is always biased. With a biased reward mean estimate, no matter what the decision-making policy is, normal agents can never find an optimal arm.

**Example 4** *Consider a 4-agent complete graph, with agent 1 being a Byzantine agent and agents 2–4 being normal agents. Let one arm $k$ be of Bernoulli distribution with mean $\frac{1}{2}$. At each time, agent 1 sends $\frac{1}{3}$ to all three normal agents as its reward mean estimate on arm $k$.*

**Lemma 1** *With Example 4, there holds $|\mathbf{E}(\tilde{z}_{i,k}(t)) - \mu_k| \geq \frac{1}{24}$ for any $i \in \{2, 3, 4\}$ and $t \in \{1, 2, \ldots\}$.*

**Proof of Lemma 1:** Since the neighbor graph is complete and $\tilde{z}_{1i,k}(t) = \frac{1}{3}$ for all $i \in \{2, 3, 4\}$, it is easy to see that

$$\mathbf{E}(\tilde{z}_{2,k}(t)) = \mathbf{E}(\tilde{z}_{3,k}(t)) = \mathbf{E}(\tilde{z}_{4,k}(t)).$$

Take agent 2 as an example, at each time $t$, it receives three pieces of reward information and only retains one of them for update. Let $p(t)$ be the probability that agent 2 retains the Byzantine value at time $t$. Then, we have

$$\mathbf{E}(\tilde{z}_{2,k}(t+1)) = \frac{1}{2}\mathbf{E}(\tilde{z}_{2,k}(t)) + \frac{1}{2}\left(\frac{1}{3}p(t) + \left(\frac{\mathbf{E}(\tilde{z}_{3,k}(t))}{2} + \frac{\mathbf{E}(\tilde{z}_{4,k}(t))}{2}\right)(1 - p(t))\right)$$

$$= \left(1 - \frac{p(t)}{2}\right)\mathbf{E}(\tilde{z}_{2,k}(t)) + \frac{1}{6}p(t).$$

If $\mathbf{E}(\tilde{z}_{2,k}(t)) \leq \frac{1}{3}$, then there holds

$$\mathbf{E}(\tilde{z}_{2,k}(t+1)) \leq \frac{1}{3}\left(1 - \frac{p(t)}{2}\right) + \frac{1}{6}p(t) = \frac{1}{3},$$

and if $\mathbf{E}(\tilde{z}_{2,k}(t)) > \frac{1}{3}$, then

$$\mathbf{E}(\tilde{z}_{2,k}(t+1)) \leq \mathbf{E}(\tilde{z}_{2,k}(t)) - \frac{1}{3}\frac{p(t)}{2} + \frac{p(t)}{6} = \mathbf{E}(\tilde{z}_{2,k}(t)).$$

Together we have

$$\mathbf{E}(\tilde{z}_{2,k}(t+1)) \leq \max\left\{\mathbf{E}(\tilde{z}_{i,k}(t)), \frac{1}{3}\right\}. \tag{9}$$

Now consider $t = 0$. It is easy to see that $\mathbf{E}(\tilde{z}_{i,k}(0)) = \mathbf{E}(X_{i,k}(0)) = \frac{1}{2}$ for $i \in \{2,3,4\}$ and that $p(0) = \frac{1}{2}$. Thus,

$$\mathbf{E}(\tilde{z}_{2,k}(1)) = \frac{11}{24}.$$

Then, from equation 9, we obtain that for any $t \in \{1, \ldots, T\}$,

$$\mathbf{E}(\tilde{z}_{2,k}(t)) \leq \frac{11}{24},$$

since $\mu_k = \frac{1}{2}$, $|\mathbf{E}(\tilde{z}_{i,k}(t)) - \mu_k| \geq \frac{1}{24}$, which completes the proof. ∎

The above Example 4 can be straightforwardly generalized to any size complete graphs using similar arguments to those in the proof of Lemma 1.

**Proposition 1** *Consider an $n$-agent complete neighbor graph with $f$ agents being Byzantine. Set one arm $k$'s reward to be of Bernoulli distribution with mean $\mu_k$ and let all $f$ Byzantine agents always transmit a contant value $\vartheta_k \neq \mu_k$ to their out-neighbors at each time. If $n \geq 4$ and $f \leq \lfloor \frac{n-1}{2} \rfloor$, then $|\mathbf{E}(\tilde{z}_{i,k}(t)) - \mu_k|$ is uniformly bounded below by a strictly positive constant for all normal agent $i$ and time $t$.*

Proposition 1 shows that running consensus can lead to biased reward mean estimates. With a biased reward mean, normal agents will possibly misidentify an optimal arm, and as a result, no matter what decision-making policy normal agents use, it may yield a linear regret. We will run experiments in Appendix G to visually demonstrate this result; see Figure 9 and its discussion.

## F    Analysis

This section provides the analysis of the algorithm and proofs of the main results in the paper.

**Lemma 2** *If $|\mathcal{A}_{i,k}(t)| > 2f$, then $z_{i,k}(t)$ in equation 3 can be expressed as a convex combination of $\bar{x}_{i,k}(t)$ and all $\bar{x}_{j,k}(t)$, $j \in \mathcal{A}_{i,k}(t) \cap \mathcal{H}$ in that*

$$z_{i,k}(t) = w_{ii,k}(t)\bar{x}_{i,k}(t) + \sum_{j \in \mathcal{A}_{i,k}(t) \cap \mathcal{H}} w_{ij,k}(t)\bar{x}_{j,k}(t), \tag{10}$$

*where $w_{ii,k}(t)$ and $w_{ij,k}(t)$ are non-negative numbers satisfying $w_{ii,k}(t) + \sum_{j \in \mathcal{N}_i \cap \mathcal{H}} w_{ij,k}(t) = 1$, and there exists a positive constant $\eta_{i,k}(t) = \frac{1}{|\mathcal{B}_{i,k}(t)|+1}$ such that for all $i \in \mathcal{H}$ and $t$, $w_{ii,k}(t) \geq \eta_{i,k}(t)$ and among all $w_{ij,k}(t)$, $k \in \mathcal{A}_{i,k}(t) \cap \mathcal{H}$, at least $|\mathcal{B}_{i,k}(t)|$ of them are bounded below by $\eta_{i,k}(t)/2$.*

**Proof of Lemma 2:** From definition, $|\mathcal{B}_{i,k}(t) \cap \mathcal{F}|$ means the number of Byzantine agents being retained by agent $i$ for arm $k$ after the two filtering steps. From Trimmed Mean Filter, having $|\mathcal{B}_{i,k}(t) \cap \mathcal{F}|$ Byzantine agents being retained indicates that there are $|\mathcal{B}_{i,k}(t) \cap \mathcal{F}|$ normal agents who are retained after Consistency

Filter are filtered out in Trimmed Mean Filter due to having a small sample mean and $|\mathcal{B}_{i,k}(t) \cap \mathcal{F}|$ are filtered out due to having a large sample mean. In this sense, for any $j \in \mathcal{B}_{i,k}(t) \cap \mathcal{F}$, there exists a distinct pair of normal agent $j^-, j^+ \in \mathcal{A}_{i,k}(t) \setminus \mathcal{B}_{i,k}(t) \cap \mathcal{H}$, such that $\bar{x}_{j^-,k}(t) \leq \bar{x}_{j,k}(t) \leq \bar{x}_{j^+,k}(t)$. Then, $\bar{x}_{j,k}(t)$ can be expressed as a convex combination of $\bar{x}_{j^-,k}(t)$ and $\bar{x}_{j^+,k}(t)$, i.e., there exists a $\beta_j$, such that

$$\bar{x}_{j,k}(t) = \beta_j \bar{x}_{j^-,k}(t) + (1 - \beta_j) \bar{x}_{j^+,k}(t).$$

Then, equation 3 can be rewritten as

$$z_{i,k}(t) = \frac{1}{|\mathcal{B}_{i,k}(t)| + 1} \left( \bar{x}_{i,k}(t) + \sum_{h \in \mathcal{B}_{i,k}(t) \cap \mathcal{H}} \bar{x}_{h,k}(t) + \sum_{j \in \mathcal{B}_{i,k}(t) \cap \mathcal{F}} (\beta_j \bar{x}_{j^-,k}(t) + (1 - \beta_j) \bar{x}_{j^+,k}(t)) \right).$$

Since $\mathcal{B}_{i,k}(t) \cap \mathcal{H} \subset \mathcal{A}_{i,k}(t) \cap \mathcal{H}$ and $j^-, j^+ \in \mathcal{A}_{i,k}(t) \cap \mathcal{H}$ for all $j \in \mathcal{B}_{i,k}(t) \cap \mathcal{F}$, the above equation is exactly in the form of equation 10. Besides, for each $j \in \mathcal{B}_{i,k}(t) \cap \mathcal{F}$, at least one of $\beta_j$ and $1 - \beta_j$ are lower bounded by $\frac{1}{2}$. Then, the number of $w_{ij,k}(t)$ that are lower bounded by $\frac{1}{2(|\mathcal{B}_{i,k}(t)|+1)}$ is at least $|\mathcal{B}_{i,k}(t) \cap \mathcal{H}| + |\mathcal{B}_{i,k}(t) \cap \mathcal{F}| = |\mathcal{B}_{i,k}(t)|$, which completes the proof. ∎

**Lemma 3** (Hoeffding's inequality (Hoeffding, 1963, Theorem 2)) *If $X_1, \ldots, X_n$ are independent random variables and $a_i \leq X_i \leq b_i$ for $i = 1, \ldots, n$, let $\bar{X} = \frac{\sum_{i=1}^n X_i}{n}$ and $\mu$ be the expectation of $\bar{X}$; then for any $\epsilon > 0$, it holds that*

$$\mathbf{P}(\bar{X} - \mu \geq \epsilon) \leq \exp\left( -\frac{2n^2 \epsilon^2}{\sum_{i=1}^n (b_i - a_i)^2} \right).$$

Now we are in a position to prove the theorems.

**Proof of Theorem 1:** The lower bound for single-agent algorithm is $\Omega(\log T)$ (Lai & Robbins, 1985, Theorem 1). For all network structure, a complete graph leads to best performance as each agent has full knowledge of the reward information over the network. For each agent, this is equivalent to a single-agent setting where it selects one arm and receives $N$ pieces of reward information at each time instance. From Filtering step B, at least $2f$ pieces of information are filtered out if $\mathcal{B}_{i,k}(t)$ is nonempty. This is to say, each agent at most retains $\max\{|\mathcal{N}_i| - 2f, 0\}$ neighbors after the whole filtering process and thus is able to make use of at most $\max\{|\mathcal{N}_i| - 2f, 1\}$ pieces of reward information in the updates, which leads to a lower bound of order $\Omega(\frac{\log T}{\max\{|\mathcal{N}_i| - 2f + 1, 1\}})$. ∎

**Proof of Theorem 2:** Since $R_i(T) = \sum_{k:\Delta_k > 0} \mathbf{E}(n_{i,k}(T)) \Delta_k$, we can convert the problem into finding an estimate of $\mathbf{E}(n_{i,k}(T))$. Let

$$L = \left\lceil \max_{t \in \{1,\ldots,T\}} \frac{8 g_{i,k}(t) \log t}{\Delta_k^2} \right\rceil$$

for any $i \in \mathcal{H}$, if $n_{i,k}(T) \geq L$, let $t_i \leq T$ be the time such that $n_{i,k}(t_i) = L$; then we have

$$n_{i,k}(T) = L + \sum_{t=t_i+1}^T \mathbb{1}(a_i(t) = k).$$

Based on this fact, we obtain that

$$n_{i,k}(T) \leq L + \sum_{t=1}^T \mathbb{1}(a_i(t) = k, n_{i,k}(t-1) \geq L).$$

Since $n_{i,k}(t-1) \leq n_{i,k}(t)$, we immediately obtain from the above inequality that

$$n_{i,k}(T) \leq L + \sum_{t=1}^T \mathbb{1}(a_i(t) = k, n_{i,k}(t) \geq L),$$

from the decision-making step, if agent $i$ selects arm $k$ at time $t$, then $z_{i,k}(t) + C(t, n_{i,k}(t)) \geq z_{i,1}(t) + C(t, n_{i,1}(t))$. Thus,

$$n_{i,k}(T) \leq L + \sum_{t=1}^{T} \mathbb{1}\Big(z_{i,k}(t) + C(t, n_{i,k}(t)) \geq z_{i,1}(t) + C(t, n_{i,1}(t)), n_{i,k}(t) \geq L\Big).$$

Since for any $t \in \{1, \ldots, T\}$, we have $1 \leq n_{i,k}(t) \leq t$ for all $i \in \mathcal{H}$ and $k \in \mathcal{M}$, considering all possible values of $n_{i,k}(t)$ and $n_{i,1}(t)$ for a given $t$ in the above inequality, we obtain that

$$n_{i,k}(T) \leq L + \sum_{t=1}^{T} \sum_{N_{ik}=L}^{t} \sum_{N_{i1}=1}^{t} \mathbb{1}\Big(z_{i,k}(t) + C(t, n_{i,k}(t)) \geq z_{i,1}(t) + C(t, n_{i,1}(t)),$$
$$n_{i,k}(t) = N_{ik}, n_{i,1}(t) = N_{i1}\Big). \tag{11}$$

Applying the expectation operation on both sides of the above inequality, it holds that

$$\mathbf{E}(n_{i,k}(T)) \leq L + \sum_{t=1}^{T} \sum_{N_{ik}=L}^{t} \sum_{N_{i1}=1}^{t} \mathbf{P}\Big(z_{i,k}(t) + C(t, n_{i,k}(t)) \geq z_{i,1}(t) + C(t, n_{i,1}(t)),$$
$$n_{i,k}(t) = N_{ik}, n_{i,1}(t) = N_{i1}\Big)$$
$$\leq L + \sum_{t=1}^{T} \sum_{N_{ik}=L}^{t} \sum_{N_{i1}=1}^{t} \Big(\mathbf{P}\big(z_{i,k}(t) - \mu_k \geq C(t, n_{i,k}(t)), n_{i,k}(t) = N_{ik}\big)$$
$$+ \mathbf{P}\big(\mu_1 - z_{i,1}(t) \geq C(t, n_{i,1}(t)), n_{i,1}(t) = N_{i1}\big)$$
$$+ \mathbf{P}\big(2C(t, n_{i,k}(t)) > \mu_1 - \mu_k), n_{i,k}(t) = N_{ik}\big)\Big). \tag{12}$$

It is easy to verify that when $n_{i,k}(t) \geq L$, for any $t \in \{1, \ldots, T\}$, it always holds true that $2C(t, n_{i,k}(t)) \leq \mu_1 - \mu_k$, which is to say,

$$\mathbf{P}\big(2C(t, n_{i,k}(t)) \geq \mu_1 - \mu_k), n_{i,k}(t) = N_{ik}\big) = 0.$$

Substituting the result to equation 12, we obtain that

$$\mathbf{E}(n_{i,k}(T)) \leq L + \sum_{t=1}^{T} \sum_{N_{ik}=L}^{t} \sum_{N_{i1}=1}^{t} \Big(\mathbf{P}\big(z_{i,k}(t) - \mu_k \geq C(t, n_{i,k}(t)), n_{i,k}(t) = N_{ik}\big)$$
$$+ \mathbf{P}\big(\mu_1 - z_{i,1}(t) \geq C(t, n_{i,1}(t)), n_{i,1}(t) = N_{i1}\big)\Big). \tag{13}$$

The two probabilities in equation 13 can be viewed as concentration bounds of the reward mean estimates. Without loss of generality, we only provide detailed estimation for $\mathbf{P}\big(z_{i,k}(t) - \mu_k \geq C(t, n_{i,k}(t)), n_{i,k}(t) = N_{ik}\big)$ in the following context, the analysis for $\mathbf{P}\big(\mu_1 - z_{i,1}(t) \geq C(t, n_{i,1}(t)), n_{i,1}(t) = N_{i1}\big)$ is exactly the same. And to this end, we need to consider the following two cases based on the number of neighbors that are retained after two filters.

**Case A:** If $\mathcal{B}_{i,k}(t)$ is empty, then $z_{i,k}(t) = \bar{x}_{i,k}(t)$, and the decision-making is exactly the same as that of single-agent UCB1 in Auer et al. (2002a). From the analysis in (Auer et al., 2002a, Theorem 1), it holds that

$$\sum_{N_{ik}=L}^{t} \sum_{N_{i1}=1}^{t} \Big(\mathbf{P}\big(z_{i,k}(t) - \mu_k \geq C(t, n_{i,k}(t)), n_{i,k}(t) = N_{ik}\big)$$
$$+ \mathbf{P}\big(\mu_1 - z_{i,1}(t) \geq C(t, n_{i,1}(t)), n_{i,1}(t) = N_{i1}\big)\Big) \leq \frac{2}{t^2}.$$

**Case B:** If $\mathcal{B}_{i,k}(t)$ is nonempty, let $\tilde{\mathcal{A}}_{i,k}(t) = \mathcal{A}_{i,k}(t) \cap \mathcal{H} \cup \{i\}$. from Lemma 2; we can write $z_{i,k}(t)$ as the convex combination of the sample mean of agents in $\tilde{\mathcal{A}}_{i,k}(t)$. Then,

$$\mathbf{P}\big(z_{i,k}(t) - \mu_k \geq C(t, n_{i,k}(t)), n_{i,k}(t) = N_{ik}\big)$$
$$= \mathbf{P}\bigg(\sum_{j \in \tilde{\mathcal{A}}_{i,k}(t)} w_{ij,k}(t)\bar{x}_{j,k}(t) - \mu_k \geq C(t, n_{i,k}(t)), n_{i,k}(t) = N_{ik}\bigg).$$

From Consistency Filter, for any $j \in \tilde{\mathcal{A}}_{i,k}(t)$, we have $\kappa_i n_{j,k}(t) \geq n_{i,k}(t)$, consider all possible values of $n_{j,k}(t)$ for $j \in \tilde{\mathcal{A}}_{i,k}(t)$, we obtain that

$$\mathbf{P}\big(z_{i,k}(t) - \mu_k \geq C(t, n_{i,k}(t)), n_{i,k}(t) = N_{ik}\big)$$
$$= \sum_{N_{j_1 k} \geq \frac{N_{ik}}{\kappa_i}} \cdots \sum_{N_{j_{|\tilde{\mathcal{A}}_{i,k}(t)|} k} \geq \frac{N_{ik}}{\kappa_i}} \mathbf{P}\bigg(\sum_{j \in \tilde{\mathcal{A}}_{i,k}(t)} w_{ij,k}(t)\bar{x}_{j,k}(t) - \mu_k \geq C(t, n_{i,k}(t)),$$
$$n_{j,k}(t) = N_{j_k} \text{ for all } j \in \tilde{\mathcal{A}}_{i,k}(t)\bigg), \tag{14}$$

where $j_1, \ldots, j_{\tilde{\mathcal{A}}_{i,k}(t)}$ are the labels of all the agents in $\mathcal{A}_{i,k}(t) \cap \mathcal{H}$. Let $\mathcal{C}(t)$ be the collection of $\{N_{jk} : j \in \tilde{\mathcal{A}}_{i,k}(t)\}$ such that $N_{jk} \geq N_{ik}/\kappa_i$ for all $j \in \tilde{\mathcal{A}}_{i,k}(t)$. Then,

$$\mathbf{P}\big(z_{i,k}(t) - \mu_k \geq C(t, n_{i,k}(t)), n_{i,k}(t) = N_{ik}\big)$$
$$\leq \max_{\mathcal{C}(t)} \mathbf{P}\bigg(\sum_{j \in \tilde{\mathcal{A}}_{i,k}(t)} w_{ij,k}(t)\bar{x}_{j,k}(t) - \mu_k \geq C(t, n_{i,k}(t)), n_{j,k}(t) = N_{jk} \text{ for all } j \in \tilde{\mathcal{A}}_{i,k}(t)\bigg).$$

Let $\bar{X}_{i,k}(s)$ denote the averaged reward of agent $i$ on arm $k$ for a fixed sampling times $s$, it is clear that $\bar{X}_{i,k}(\cdot)$ for all $i \in \mathcal{H}$ are independent. In this sense, we have

$$\mathbf{P}\big(z_{i,k}(t) - \mu_k \geq C(t, n_{i,k}(t)), n_{i,k}(t) = N_{ik}\big)$$
$$\leq \max_{\mathcal{C}(t)} \mathbf{P}\bigg(\sum_{j \in \tilde{\mathcal{A}}_{i,k}(t)} w_{ij,k}(t)\bar{X}_{j,k}(N_{jk}) - \mu_k \geq C(t, N_{ik})\bigg).$$

Then, using Lemma 3,

$$\mathbf{P}\big(z_{i,k}(t) - \mu_k \geq C(t, n_{i,k}(t)), n_{i,k}(t) = N_{ik}\big) \leq \max_{\mathcal{C}(t)} \exp\bigg(-\frac{C^2(t, N_{ik})}{\sum_{j \in \tilde{\mathcal{A}}_{i,k}(t)} (w_{ij,k}(t))^2 \frac{1}{2N_{jk}}}\bigg) \tag{15}$$

Since $N_{jk} \geq N_{ik}/\kappa_i$ for all $j \in \mathcal{A}_{i,k}(t) \cap \mathcal{H}$, we have

$$\mathbf{P}\big(z_{i,k}(t) - \mu_k \geq C(t, n_{i,k}(t)), n_{i,k}(t) = N_{ik}\big)$$
$$\leq \exp\bigg(-\frac{2C^2(t, N_{ik})N_{ik}}{(w_{ii,k}(t))^2 + \kappa_i \sum_{j \in \mathcal{A}_{i,k}(t) \cap \mathcal{H}} (w_{ij,k}(t))^2}\bigg). \tag{16}$$

From Lemma 2, at least $|\mathcal{B}_{i,k}(t)|$ of $w_{ij,k}(t)$ for $j \in \mathcal{A}_{i,k}(t) \cap \mathcal{H}$ are lower bounded by $\eta_{i,k}(t)/2$ and that $w_{ii,k}(t) + \sum_{j \in \mathcal{A}_{i,k}(t) \cap \mathcal{H}} w_{ij,k}(t) = 1$. Thus,

$$(w_{ii,k}(t))^2 + \kappa_i \sum_{j \in \mathcal{A}_{i,k}(t) \cap \mathcal{H}} (w_{ij,k}(t))^2$$
$$\leq \eta_{i,k}^2(t) + \kappa_i\Big((|\mathcal{B}_{i,k}(t)| - 1)(\eta_{i,k}(t)/2)^2 + (1 - \eta_{i,k}(t) - (|\mathcal{B}_{i,k}(t)| - 1)\eta_{i,k}(t)/2)^2\Big)$$
$$\leq \eta_{i,k}^2(t) + \kappa_i\Big(\frac{\eta_{i,k}(t)}{4} + \frac{1}{4}\Big) = g_{i,k}(t). \tag{17}$$

Substituting equation 17 to equation 16, we obtain that

$$\mathbf{P}\big(z_{i,k}(t) - \mu_k \geq C(t, n_{i,k}(t)), n_{i,k}(t) = N_{ik}\big) \leq \frac{1}{t^4}.$$

Similarly,

$$\mathbf{P}\big(\mu_1 - z_{i,1}(t) \geq C(t, n_{i,1}(t)), n_{i,1}(t) = N_{i1}\big) \leq \frac{1}{t^4}.$$

Then, combining the above two inequalities together,

$$\sum_{N_{ik}=L}^{t} \sum_{N_{i1}=1}^{t} \left( \mathbf{P}\big(z_{i,k}(t) - \mu_k \geq C(t, n_{i,k}(t)), n_{i,k}(t) = N_{ik}\big) \right.$$
$$\left. + \mathbf{P}\big(\mu_1 - z_{i,1}(t) \geq C(t, n_{i,1}(t)), n_{i,1}(t) = N_{i1}\big) \right) \leq \frac{2}{t^2}.$$

Combining the results of both Case A and Case B with equation 13, we obtain that

$$\mathbf{E}(n_{i,k}(T)) \leq L + \sum_{t=1}^{T} \frac{2}{t^2} \leq L + \frac{\pi^2}{3}, \tag{18}$$

and further

$$R_i(T) = \sum_{k:\Delta_k>0} \mathbf{E}(n_{i,k}(T))\Delta_k \leq \sum_{k:\Delta_k>0} \left( \max_{t\in\{1,\ldots,T\}} \frac{8g_{i,k}(t)\log t}{\Delta_k} + \left(\frac{\pi^2}{3}+1\right)\Delta_k \right).$$

It is easy to see that $\tau = 1, \ldots, T$, it holds that

$$R_i(T) \leq R_i(\tau) + (T-\tau)\max_{k\in\mathcal{M}} \Delta_k,$$

where $R_i(\tau)$ denotes the expected cumulative regret at time $\tau$, and $(T-\tau)\max_{k\in\mathcal{M}} \Delta_k$ is the maximal reward loss gained after $\tau$ by keeping selecting the arm with the lowest reward mean. In this sense, we can obtain the following tighter upper bound for the reward mean estimate,

$$R_i(T) \leq \min_{\tau\in\{1,\ldots,T\}} \big( R_i(\tau) + (T-\tau)\max_{k\in\mathcal{M}} \Delta_k \big)$$
$$\leq \min_{\tau\in\{1,\ldots,T\}} \left( \sum_{k:\ \Delta_k>0} \left( \max_{t\in\{1,\ldots,\tau\}} \frac{8g_{i,k}(t)\log t}{\Delta_k} + \left(1+\frac{\pi^2}{3}\right)\Delta_k \right) + (T-\tau)\Delta_M \right),$$

which completes the proof as $\max_{k\in\mathcal{M}} \Delta_k = \Delta_M$. ∎

**Proof of Theorem 3:** We only need to show for any $t \in \{1, \ldots, T\}, i \in \mathcal{H}$ and $k \in \mathcal{M}$, it holds that $g_{i,k}(t) \leq 1$. Consider the case when $\mathcal{B}_{i,k}(t)$ is nonempty, i.e., $|\mathcal{B}_{i,k}(t)| \geq 1$. Then,

$$g_{i,k}(t) = \frac{\kappa_i|\mathcal{B}_{i,k}(t)|^2 + 3\kappa_i|\mathcal{B}_{i,k}(t)| + 2\kappa_i + 4}{4(|\mathcal{B}_{i,k}(t)|+1)^2}$$
$$= \frac{\kappa_i + \frac{\kappa_i}{|\mathcal{B}_{i,k}(t)|+1} + \frac{4}{(|\mathcal{B}_{i,k}(t)|+1)^2}}{4}$$
$$\leq \frac{\kappa_i + \frac{\kappa_i}{2} + 1}{4} < 1. \tag{19}$$

Note that $g_{i,k}(t) = 1$ when $\mathcal{B}_{i,k}(t)$ is empty. We thus can conclude that $g_{i,k}(t) \leq 1$, which completes the proof. ∎

**Proof of Theorem 4:** We only need to show that for any $t > 0$ and $k \in \mathcal{M}$, there exists an agent $i_k(t)$ such that its regret at time $t$ is strictly lower than that of the single-agent case, i.e.,

$$R_{i_k(t)}(t) < \sum_{k:\ \Delta_k>0} \left( \frac{8\log T}{\Delta_k} + \left(1+\frac{\pi^2}{3}\right)\Delta_k \right),$$

and to this end, from Theorem 2, we only need to show that $g_{i_k(t),k}(t) < 1$.

For any $k \in \mathcal{M}$, let $i_k(t) = \arg\min_{i \in \mathcal{H}} n_{i,k}(t)$. From the algorithm, agent $i_k(t)$ does not filter out any normal neighbor in Filtering Step A, as for all $j \in \mathcal{N}_{i_k(t)} \cap \mathcal{H}$, it holds that $\kappa_i n_{j,k}(t) \geq n_{j,k}(t) \geq n_{i,k}(t)$. Then, agent $i_k(t)$ at most filter out $f$ (Byzantine) agents at time $t$ in Filtering Step A. As a result, there holds $|\mathcal{A}_{i_k(t),k}(t)| \geq 2f + 1$ and thus $|\mathcal{B}_{i_k(t),k}(t)| \geq 1$. Then, from equation 19,

$$g_{i_k(t),k}(t) < 1.$$

From Theorem 3, we have $g_{i_k(t),k}(t) \leq 1$ for all $i \in \mathcal{H}$. Then, it holds for any $t > 0$ that

$$\sum_{i \in \mathcal{H}} g_{i,k}(t) < |\mathcal{H}|. \tag{20}$$

From Theorem 2, $R(T)$ satisfies

$$R(T) \leq \sum_{i \in \mathcal{H}} \sum_{k:\ \Delta_k > 0} \left( \max_{t \in \{1,\ldots,T\}} \frac{8 g_{i,k}(t) \log t}{\Delta_k} + \left(\frac{\pi^2}{3} + 1\right)\Delta_k \right)$$
$$= \sum_{k:\ \Delta_k > 0} \left( \max_{t \in \{1,\ldots,T\}} \frac{8 \sum_{i \in \mathcal{H}} g_{i,k}(t) \log t}{\Delta_k} + \left(\frac{\pi^2}{3} + 1\right)|\mathcal{H}|\Delta_k \right).$$

Then, using equation 20, there holds

$$R(T) < |\mathcal{H}| \sum_{k:\ \Delta_k > 0} \left( \max_{t \in \{1,\ldots,T\}} \frac{8 \log t}{\Delta_k} + \left(\frac{\pi^2}{3} + 1\right)\Delta_k \right)$$
$$= |\mathcal{H}| \sum_{k:\ \Delta_k > 0} \left( \frac{8 \log T}{\Delta_k} + \left(\frac{\pi^2}{3} + 1\right)\Delta_k \right),$$

which completes the proof. ∎

**Proof of Theorem 5:** From Theorem 2, the network total regret $R(T)$ satisfies

$$R(T) \leq \sum_{k:\ \Delta_k > 0} \left( \max_{t \in \{1,\ldots,T\}} \frac{8 \sum_{i \in \mathcal{H}} g_{i,k}(t) \log t}{\Delta_k} + \left(\frac{\pi^2}{3} + 1\right)|\mathcal{H}|\Delta_k \right). \tag{21}$$

From equation 19, when the $3f + 1$ degree requirement is satisfied at time $t$ for any $t = 1, \ldots, T$,

$$\sum_{i \in \mathcal{H}} g_{i,k}(t) < |\mathcal{H}|,$$

and when it is not satisfied, we have a weaker result

$$\sum_{i \in \mathcal{H}} g_{i,k}(t) \leq |\mathcal{H}|.$$

We discuss in the following analysis all the possibilities of the "last" time that the $3f + 1$ degree requirement is not satisfied.

If the last time that the degree requirement is not met is $T$, then $g_{i,k}(T) = 1$ and we have

$$\max_{t \in \{1,\ldots,T\}} \sum_{i \in \mathcal{H}} g_{i,k}(t) \log t \leq |\mathcal{H}| \log T.$$

From the random graph structure, this situation happens with probability $1 - p$. If the last time that the degree requirement is not met is $T - 1$, then we have

$$\max_{t \in \{1,\ldots,T\}} \sum_{i \in \mathcal{H}} g_{i,k}(t) \log t \leq \max \left\{ |\mathcal{H}| \log(T - 1), \sum_{i \in \mathcal{H}} g_{i,k}(T) \log T \right\},$$

and this happens with probability $(1-p)p$. Continuing this process, If the last time that the degree requirement is not met is $T - \tau$, where $0 \leq \tau < T$, then we have

$$\max_{t \in \{1,\ldots,T\}} \sum_{i \in \mathcal{H}} g_{i,k}(t) \log t$$

$$\leq \max\left\{|\mathcal{H}| \log(T - \tau), \sum_{i \in \mathcal{H}} g_{i,k}(t) \log t \text{ for } t = T - \tau + 1, \ldots, T\right\},$$

and this happens with probability $(1-p)p^\tau$. And if the degree requirement is always met, then

$$\max_{t \in \{1,\ldots,T\}} \sum_{i \in \mathcal{H}} g_{i,k}(t) \log t \leq \max\left\{\sum_{i \in \mathcal{H}} g_{i,k}(t) \log t \text{ for } t = 1, \ldots, T\right\},$$

and this happens with probability $p^T$. For $0 \leq \tau \leq T$, let

$$h(\tau, T) = \max\left\{|\mathcal{H}| \log(\max\{1, T - \tau\}), \sum_{i \in \mathcal{H}} g_{i,k}(t) \log t \text{ for } t = T - \tau + 1, \ldots, T\right\}.$$

Then, from equation 21,

$$R(T) \leq \sum_{k:\, \Delta_k > 0} \left(\frac{8 \sum_{\tau=0}^{T-1} (1-p)p^\tau \mathbf{E}(h(\tau, T)) + 8p^T \mathbf{E}(h(T, T))}{\Delta_k} + \left(\frac{\pi^2}{3} + 1\right)|\mathcal{H}|\Delta_k\right). \qquad (22)$$

Since from the proof of Theorem 3,

$$h(T, T) = \max\left\{\sum_{i \in \mathcal{H}} g_{i,k}(t) \log t \text{ for } t = 1, \ldots, T\right\} < \sum_{i \in \mathcal{H}} \max_{t \in \{1,\ldots,T\}} \log t = |\mathcal{H}| \log T,$$

and $h(\tau, T) \leq |\mathcal{H}| \log T$ for $0 \leq \tau \leq T - 1$, then using equation 22,

$$R(T) < \sum_{k:\, \Delta_k > 0} \left(\frac{8 \sum_{\tau=0}^{T-1} (1-p)p^\tau + 8p^T}{\Delta_k}|\mathcal{H}| \log T + \left(\frac{\pi^2}{3} + 1\right)|\mathcal{H}|\Delta_k\right)$$

$$= |\mathcal{H}| \sum_{k:\, \Delta_k > 0} \left(\frac{8 \log T}{\Delta_k} + \left(\frac{\pi^2}{3} + 1\right)\Delta_k\right).$$

The right hand side is the network total regret bound of the non-cooperative counterpart. Thus, we conclude that our regret bound is always strictly better than the non-cooperative counterpart. Moreover, let $\Phi(T)$ be the difference in upper bound between the two algorithms, i.e.,

$$\Phi(T) = \sum_{k:\, \Delta_k > 0} \left(\frac{8|\mathcal{H}| \log T}{\Delta_k} - \frac{8 \sum_{\tau=0}^{T-1} (1-p)p^\tau \mathbf{E}(h(\tau, T)) + 8p^T \mathbf{E}(h(T, T))}{\Delta_k}\right)$$

$$= 8 \sum_{k:\, \Delta_k > 0} \frac{\sum_{\tau=1}^{T-1} (1-p)p^\tau (|\mathcal{H}| \log T - \mathbf{E}(h(\tau, T))) + p^T (|\mathcal{H}| \log T - \mathbf{E}(h(T, T)))}{\Delta_k}.$$

Since for any $\tau = 1, \ldots, T$, we have

$$\mathbf{E}(h(\tau, T)) = \mathbf{E}\left(\max\left\{|\mathcal{H}| \log(\max\{1, T - \tau\}), \sum_{i \in \mathcal{H}} g_{i,k}(t) \log t \text{ for } t = T - \tau + 1, \ldots, T\right\}\right)$$

$$\geq \mathbf{E}(|\mathcal{H}| \log(\max\{1, T - \tau\})) = |\mathcal{H}| \log(\max\{1, T - \tau\}),$$

substituting it to the definition of $\Phi(T)$, we obtain that

$$
\begin{aligned}
\Phi(T) &\leq 8|\mathcal{H}| \sum_{k:\, \Delta_k > 0} \frac{\sum_{\tau=1}^{T-1}(1-p)p^\tau \log \frac{T}{T-\tau} + p^T \log T}{\Delta_k} \\
&= 8|\mathcal{H}| \sum_{k:\, \Delta_k > 0} \frac{\left(\sum_{\tau=1}^{T/2} + \sum_{\tau=T/2+1}^{T-1}\right)(1-p)p^\tau \log \frac{T}{T-\tau} + p^T \log T}{\Delta_k} \\
&\leq 8|\mathcal{H}| \sum_{k:\, \Delta_k > 0} \frac{\sum_{\tau=1}^{T/2}(1-p)p^\tau \log 2 + \sum_{\tau=T/2+1}^{T-1}(1-p)p^\tau \log T + p^T \log T}{\Delta_k} \\
&\leq 8|\mathcal{H}| \sum_{k:\, \Delta_k > 0} \frac{p \log 2 + (1-p)p^{\frac{T}{2}} T \log T/2 + p^T \log T}{\Delta_k},
\end{aligned}
$$

since for $p \in (0,1)$, both $(1-p)p^{\frac{T}{2}}T\log T$ and $p^T \log T$ are of order $o(1)$, we conclude that $\Phi(T) \leq O(1)$, which completes the proof. ∎

## G  Additional Simulations

We begin with the simulation with the same arm setting and Byzantine policy as in Section 5 to further validate the theoretical results. For all simulations in this section, the total time $T$ is chosen to be 10000, and the figures are the averaged result of 50 runs.

We first consider a ten-agent network with two of the agents being Byzantine. The graphs change over time but always satisfies that $f = 1$. We consider two cases: (1) Each normal agent has at least $4 = 3f + 1$ neighbors and the average number of neighbor of each normal agent equals 4.5. (2) Each directed edge is activated with a common probability $q = 1/2$. For both cases, the average in-degree for each normal agent equals 4.5. The simulation result for the two-case comparison is given in Figure 8. Observe that the performance of the two cases differs significantly despite having a same average in-degree for each normal agent. This is because case (1) guarantees the $3f + 1$ degree requirement be satisfied at each time whereas case (2) does not. This observation is consistent with Remark 3.

Next we consider a ten-agent complete network with one agent being Byzantine. Since the updating processes in Landgrena et al. (2021); Landgren et al. (2016); Martínez-Rubio et al. (2019) are alike, we only take Landgrena et al. (2021) as an example. We compare the performance of our algorithm with Zhu & Liu (2021) with a trimmed mean filter (denoted as Algorithm 3 in Figure 9) and Landgrena et al. (2021) with a trimmed mean filter (denoted as Algorithm 4 in Figure 9) . For Zhu & Liu (2021), we use the same Byzantine policy as ours, and for Landgrena et al. (2021), the Byzantine agent follows a similar rule: it sets $[0.4, 0.5, 0.4, 0.3]$ as $r$ information for each arm, randomly copies one normal neighbor's $\hat{n}$, and sets $\hat{s} = \hat{n} * r$, and then broadcasts these variables to its neighbors. The definition of $r, \hat{n}, \hat{s}$ can be found in Landgrena et al. (2021). The simulation result is shown in Figure 9. It is clearly shown that the algorithms in Zhu & Liu (2021); Landgrena et al. (2021) are not resilient even with a Byzantine filter and over a complete graph, as their corresponding regret curves appear to be linear, this observation is consistent with Lemma 1.

### G.1  Application in Recommender Systems

In this appendix, we consider the application in the recommender systems, where an advertising firm is charged with the task of presenting one type of advertisement to a user. If the user clicks the ad, the firm receives 1 as a reward, and 0-reward otherwise. Each user has individual preference and thus has different probabilities to make a click on different types of ads. Consider that ten advertising firms are seeking to present the optimal option to a user over twenty different types of ads. Each firm has connections with some others so that they can cooperate with each other for better performance. The connections form an undirected graph shown in Figure 10, with the white nodes representing for the firms that transmit honest information, and the grey nodes representing for the "evil" firms that transmit misleading information for

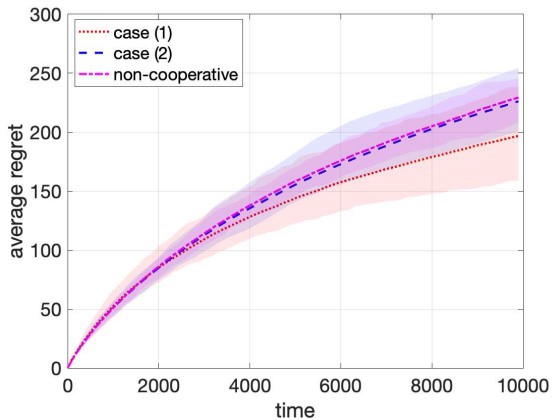

Figure 8: Simulation result of the average regret of normal agents

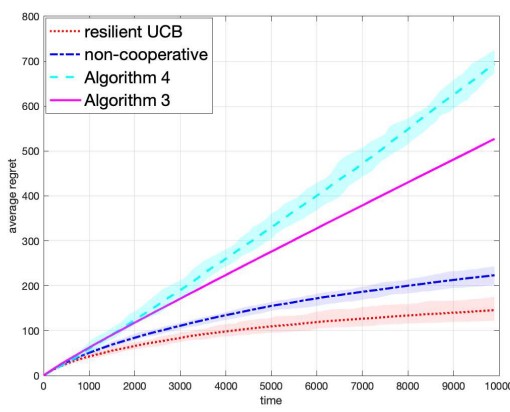

Figure 9: Simulation result of the average regret of normal agents

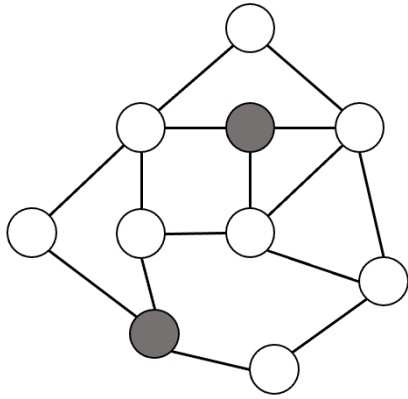

Figure 10: Connections of the ten firms

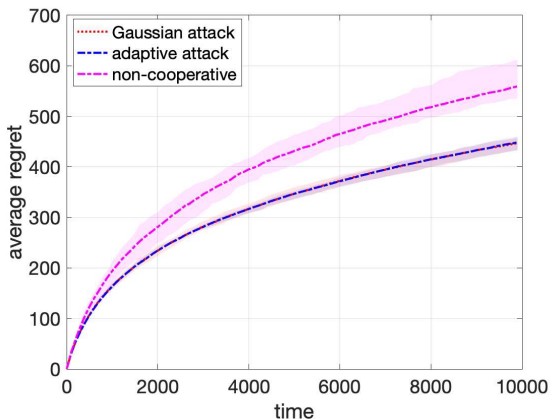

Figure 11: Simulation result of the average regret of normal agents

malicious competition. The honest firms perceive that something is amiss but cannot identify the adversarial firms so they decide to defend with a resilient decision-making. For the adversarial firms, we consider the following two Byzantine attack models.

**Gaussian Attack:** Let each adversarial firm $i$ randomly generate a bias value $\beta_{i,k} \in (0,1)$ for each arm $k$. For each neighbor $j$ and time $t$, it generates a $c_j(t)$ from $N(\beta_k, 0.01)^1$ and truncates $\bar{x}_{i,k}(t) + c_j(t)$ to the range $[0,1]$, then transmits the value to its neighbors. For sample count information at time $t$, let each adversarial firm copy $n_{j,k}(t-1)$ from a neighbor $j$.

**Adaptive Attack:** Here we assume the adversarial firms literally know everything, e.g., the arm information, the communication activities and even the Byzantine-defending algorithm. This is possible via illegitimately monitoring other firms and when they already have the user's information in their database. For each normal neighbor $i$ and an arm $k$, let an adversarial firm transmit a value slightly larger than $n_{i,k}(t)$ as the sample count information; if the arm is the optimal arm 1, let the adversarial firm transmit the second smallest

---

[1]We use $N(s, d)$ to denote the Gaussian distribution with mean $s$ and variance $d$.

$\bar{x}_{j,1}(t)$ for $j \in \mathcal{A}_{i,1}(t)$ (they are able to accurately predict $\mathcal{A}_{i,k}(t)$ via monitoring) and otherwise transmit the second largest $\bar{x}_{j,k}(t)$ for $j \in \mathcal{A}_{i,k}(t)$. In this sense, the adversarial firms are able to down-weight the optimal arm and up-weight the sub-optimal arms in the decision-making of each normal neighbor to the maximum extent.

For the simulation under the above application model, the reward distribution of each arm is set to be a Bernoulli distribution with a randomly generated mean. Each honest firm $i$ sets a $\kappa_i$ that is randomly picked from $[1, 2)$. The simulation result is shown in Figure 11. We observe that for both Byzantine attacking strategies, our algorithm ensures good performance. Besides, the performance appears to be not sensitive to different Byzantine policies as the difference of performance under the two types of attack is negligible.

Considering that the recent study Bayati et al. (2020) shows that the greedy algorithm performs extremely well when the number of arms are large, we also test on *greedy* algorithm incorporating **Filter**$(i, k, t)$, where each normal agent $i$ always selects the arm with largest $z_{i,k}(t)$ in the decision-making step, i.e., $a_i(t+1) = \arg \max_{k \in \mathcal{M}} z_{i,k}(t)$, after executing **Filter**$(i, k, t)$. We compare its performance under the adaptive Byzantine attack with algorithms (a) each normal agent $i$ runs **Filter**$(i, k, t)$, then executes softmax on the arms corresponding to the three largest $z_{i,k}(t)$ and samples one of them according to the probabilities softmax returns; (b) Resilient Decentralized UCB; (c) Resilient Decentralized UCB with a tuned parameter: similar to what the single-agent UCB Auer et al. (2002a) does in simulation, we tune the confidence parameter $C(t, n_{i,k}(t))$ to

$$C(t, n_{i,k}(t)) = \sqrt{\frac{g_{i,k}(t) \log t}{4n_{i,k}(t)}}. \tag{23}$$

The simulation results are shown in Figures 12 and 13. We observe that while softmax generally provides a linear regret possibly because agents have a probability of order $O(1)$ to select a sub-optimal arm at each time, greedy algorithm works well with **Filter**$(i, k, t)$ in resilient setting. This indicates that our filtering and updating steps (i.e. Algorithm 2) may be compatible to various classic bandit algorithms and thus have great potentials in solving bandit problems, which deserves further investigation in the future. Moreover, when $k = 20$, i.e., the arm number is large, resilient greedy algorithm has a substantial advantage over Resilient Decentralized UCB while has a notable disadvantage when dealing with a relatively small arm number, which is consistent with the observations in the non-resilient setting Bayati et al. (2020). Besides, we find that the tuned Resilient Decentralized UCB always performs significantly better than Resilient Decentralized UCB (and in fact better than the resilient greedy algorithm for both arm settings), which is consistent with the observations for single-agent UCB under the non-resilient setting Auer et al. (2002a). However, like Auer et al. (2002a), we are not able to provide a theoretical prove for the regret bound. To help see the difference of exploration procedure of the two UCB algorithms, we further divide the total time $T$ into three stages with equal length and provide the simulation of the frequency of arm selecting for normal agents when $k = 20$; see Figures 14 and 15.

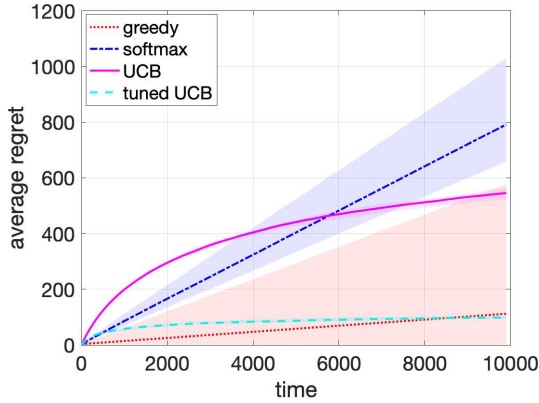

Figure 12: Simulation result of the average regret of normal agents for different resilient algorithms when $k = 20$

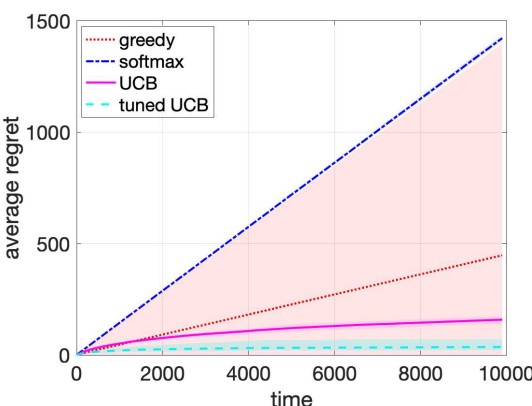

Figure 13: Simulation result of the average regret of normal agents for different resilient algorithms when $k = 4$

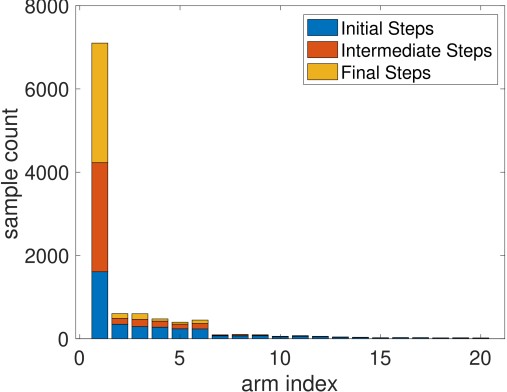

Figure 14: Simulation result of the average explorations of normal agents under Resilient Decentralized UCB

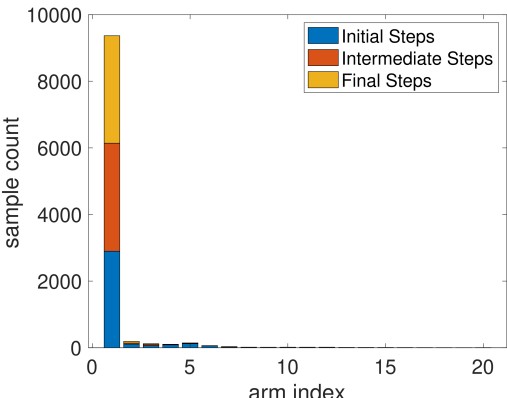

Figure 15: Simulation result of the average explorations of normal agents under **tuned** Resilient Decentralized UCB equation 23

