# OpenReview forum: "Byzantine-Resilient Decentralized Multi-Armed Bandits"
_TMLR — Accepted by TMLR_

### Review · Reviewer_ztqz · 2024-04-08

**Summary Of Contributions:**

This paper studies how to recover such salient behavior when an unknown fraction of the agents can be Byzantine, that is, communicate arbitrarily wrong information in the form of reward mean-estimates or confidence sets. This formulation is applicable to computer networks, recommender systems, and financial markets. One key contribution of this paper is the development of a fully decentralized resilient upper confidence bound (UCB) algorithm, which fuses an information mixing step among agents with a truncation of inconsistent and extreme values. This truncation step enables the algorithm to achieve the results that the regret of each normal agent is no worse than the classic single-agent UCB1 algorithm, and furthermore, the cumulative regret of all normal agents is strictly better than the non-cooperative case, provided that each agent has at least $3f + 1$ neighbors where $f$ is the maximum possible Byzantine agents in each agent’s neighborhood. The authors also provide a lower bound for this problem, and investigate the extensions to time-varying neighbor graphs. The experimental results corroborate the effectiveness of the proposed algorithm in practice.

**Audience:**

Yes

**Broader Impact Concerns:**

I do not think that this paper involves ethical concerns.

**Claims And Evidence:**

Yes

**Requested Changes:**

Please see the review on weaknesses above.

**Strengths And Weaknesses:**

**Strengths:**

1.	The studied problem, Byzantine-Resilient Decentralized Multi-Armed Bandits, is an interesting problem and finds various applications in computer networks, recommender systems, and financial markets with attacks.
2.	The authors design a fully decentralized resilient upper confidence bound (UCB) algorithm, which adopts a crucial step to truncate inconsistent and extreme values. The idea of this truncation step is interesting.
3.	The authors provide regret upper bounds to show that the regret of each normal agent is no higher than the classic single-agent UCB1 algorithm, and the cumulative regret of all normal agents is strictly better than the non-cooperative case.
4.	Empirical evaluations are conducted to demonstrate the effectiveness of the proposed algorithm.


**Weaknesses:**

1.	In Theorem 2, the factor $g_{i,k}$ seems to depend on the algorithm? Is it of constant order? It would be better to give a universal upper bound for this factor.
2.	In Theorem 2, it is unclear how the numbers of agents and Byzantine agents ($f$) influence the regret bound. In many prior cooperative multi-player bandit works, when the number of agents increases, the average regret of each agent decreases. Does the result in this paper have the similar property?
3.	It would be more clear to put a formal regret upper bound in Theorem 4, instead of just showing the relationship to the non-cooperative result.
4.	In experiments, can the authors show how the numbers of agents and Byzantine agents ($f$) influence the regret performance of the proposed algorithm in practice?

---

> ### Author Response · Authors · 2024-06-22
> **Response to Reviewer ztqz**
>
> We thank the reviewer for positive comments and valuable suggestions. The response can be summarized as follows.
>
> **Factor $g_{i,k}(t)$:** The definition of $g_{i,k}(t)$ can be found in eq. (4), it is a factor dependent on the degree (number of neighbors) of each node in the network and the number of Byzantine agents. It is of constant order and satisfies that $g_{i,k}(t)<1.$
>
> **Factors that influence regret bounds:** Our regret bound is increasing in terms of $f,$ this is because the regret is decreasing in terms of $|\scr B_{i,k}(t)|$ (according to eq. (4)) and $|\scr B_{i,k}(t)|$ is decreasing in terms of $f$ (according to the trimmed-mean filter).
>
> As for graph size $N,$ the regret bound $R_i(T)$ we derive primarily relies on the local attribute of each agent $i$, specifically, agent $i$'s local degree, rather than the scale of the network $N$. This is because throughout the algorithm, the agent operates independently of any global information, such as $N$, and hence does not require knowledge of it.  As per the definition of $g_{i,k}(t)$ in Theorem 2, generally, the regret bound tends to decrease as an agent's number of neighbors increases. This property is mentioned right after the statement of Theorem 2 and is reflected in Figure 3 (the larger $q$ is, the more averaged number of neighbors per-agent has).
>
> However, when considering a random neighbor structure where the probability each directed edge is activated is a common value $q.$ Since the expected (or averaged) local degree is determined by the graph size $N$, the regret bound does have a connection with both graph size $N$ and the number of Byzantine agents $f$. We have added a simulation accordingly, see Figures 5-6.
>
> **Formal regret bound:** The formal regret bound is provided in Theorem 2. Theorems 3 and 4 serve solely as comparative results, contrasting our algorithm with its non-cooperative counterpart.

---

### Review · Reviewer_8dob · 2024-04-15

**Summary Of Contributions:**

The paper considers decentralized stochastic multi-armed bandit in the presence of possible malevolent (Byzantine) peers. A variant of the UCB1 algorithm is proposed adapted to the multi-agent scenario. In each round agents communicate their empirical mean and sample count for each arm to their neighbors. The incoming stats are filtered out if the sample count is small. The highest and lowest mean estimations are also discarded. The remaining (unfiltered) mean estimations are mixed with the local estimations of the agent. The UCB1 bonus term is modified slightly to take into account the excess number of samples. Theoretical results show that the regret is not worse than the regret without communication, and slightly better by some constant factor in certain cases. Numerical evaluations on synthetic problems show that the proposed algorithm achieves lower regret than the non-cooperative UCB1 variant.

**Audience:**

Yes

**Claims And Evidence:**

Yes

**Requested Changes:**

There are some minor typos:
      * Algorithm 1, line 3: $n_{i,k}$ should be instead of $n_{ii,k}$
      * there is a strange symbol for the empty set in Equation (2)

In the experiments it would be useful to plot the regret bounds.

**Strengths And Weaknesses:**

The paper is well written and easy to follow.

The theoretical derivations are fairly standard for UCB-like algorithms. Having regret no worse than the non-cooperative variant is nice. The improvement on non-cooperative UCB1 is by a small constant factor, which is underwhelming given that the number of samples that can potentially be used is large. It is unclear to me that one cannot devise algorithm that achieves smaller regret bounds in the presence of Byzantine agents, or if the proposed variant is too conservative. There is an asymptotic lower bound that takes into account the graph properties and the upper bound on the malevolent neighbors, but it is difficult to match the lower and upper regret bounds.

The empirical results do confirm that the proposed algorithm is performing better than the non-cooperative UCB1 variant, but the improvement is indeed by a small factor. Ultimately, this is not surprising, since the amount of exploration driven by the bonus term is reduced by only a small factor.

---

> ### Author Response · Authors · 2024-06-22
> **Response to Reviewer 8dob**
>
> We thank the reviewer for positive comments and valuable suggestions. We have corrected the typos the reviewer raised. The other response can be summarized as follows.
>
> **Improvement over single-agent result:** Our Theorem 2 provides a detailed expression for the regret upper bound, which has a lower $\log T$ term compared with that for the single-agent counterpart. Thus, unlike the reviewer said, the regret upper bound under fixed neighboring graph (which is most considered in literature) has an improvement on the coefficient of the $\log T$ term, that is, the improvement is $O(\log T)$. Note that [Lai \& Robbins, 1985] provides a lower bound for the bandit problem, which is of order $\Omega(\log T),$ this is to say, the regret is at least $O(\log T)$ and any improvement made on the regret upper bound is at most $O(\log T).$
>
> In the presence of Byzantine agents, since they can transmit arbitrary reward information to the normal agents, without a filtering process, the reward mean estimate of the normal agents can be completely biased, leading to linearly escalating regret, let alone better than the single-agent counterpart. And even with a filtering process, the updating policy in literature is still unable to deal with this resilient problem, see discussions in Appendix E and Figure 9.
>
> **Gap between upper and lower bounds:** Firstly, despite the extensive history of bandit studies, even within the traditional single-agent framework, there remains a persistent gap between the upper and lower bounds. This discrepancy is likely inherent to bandit problems. Moving towards our resilient setting, the existence of the Byzantine agents can largely degrade the normal agents' performance. However, it is extremely difficult to analyze an algorithm-independent ``minimax'' affect the Byzantine agents can bring to the regret of the normal agents, thus in the analysis of the lower bound, we are only able to provide a relatively loose bound that does not take the Byzantine affect into account. That is why the upper and lower bounds do not match.
>
> **Plotting regret bounds:** We have added a simulation accordingly, see Figure 7. We did not add derived theoretical upper bounds in other figure plots as they each contain quite a few curves, and doing so would make the plots too crowded to be clearly visible.

---

### Review · Reviewer_2sdX · 2024-06-13

**Summary Of Contributions:**

This work studies a decentralized multi-armed bandit problem where agents employ UCB and consensus-based algorithms and have at most (known) $f$ number of byzantine in-neighbors. The authors design a message-filter algorithm (Algorithm 1) which, for each arm $k$, filters out a message if 1) its number of arm pulls is too small and 2) its reward-mean estimate is the $f$ largest or $f$ smallest among all messages. This paper then provides a byzantine-resilient decentralized multi-armed bandit algorithm that does not depend on connectivity knowledge by combining this message-filter algorithm with UCB and consensus procedures. Regret upper and lower bounds are given to show that normal agents enjoy better regret performance than their non-cooperative counterparts. This work also numerically studies the performance of the proposed algorithm.

**Audience:**

Yes

**Broader Impact Concerns:**

This is a theoretical work. The reviewer is not aware of any foreseeable broader impact.

**Claims And Evidence:**

No

**Requested Changes:**

1. While the (ii) trimmed mean technique is key to achieving resilience, the reviewer cannot understand why (i) consistency filter (Step A) is necessary. Why do the sample times need to be consistent? Is it because estimator (3) does not de-biase the unbalanced sample times of each mean? The reviewer suggests that the authors add an explanation for that.

2. The theoretical results are presented in a very informal manner. For example, the lower bound (Theorem 1) requires additional assumptions for policy (e.g., assume the policy is consistent), which are missed in the statement. Also, the equation in Theorem 1 has a redundant $\log T$ on the RHS. Theorem 3 only shows a regret upper bound for a normal agent. As the RHS is also an upper bound, the theorem cannot infer that the normal agent with cooperation outperforms non-cooperative UCB1. Similar inference issues appear in Theorems 4 and 5. The "strictly better" statement is not well supported.

3. In the proof of Theorem 3, the last inequality of equation (17) holds only if truncation parameter $\kappa_i < \frac{4}{3}$; however, the value of $\kappa_i$ is allowed to be in $[1, 2)$, as introduced in section 3. It seems this makes Theorem 3 invalid. Please clarify that.

4. Regret upper bound of the proposed algorithm (Theorem 2) does not match the given regret lower bound (Theorem 1). Among the regret upper and lower bounds, which is tight and which is loose? For the loose bound(s), please discuss the difficulties of tightening it. Moreover, the proof of Theorem 1 lacks details.

5. In Appendix E, on page 17, the notation $z_{i, k}(t+1)$ is abused to denote both the reward-mean estimate in Zhu \& Liu (2021) as well as the reward-mean estimate in this work; this makes Lemma 1 confusing. Moreover, the first equation in Appendix E makes use of all messages to update the reward-mean estimate; however, the proof of Lemma 1 assumes only one message is retained for the update.

6. Besides the only limitation (extension to the contextual setting) mentioned in the conclusion section, other limitations include that the regret is not monotone in terms of $\kappa_i$ (which means the practitioners need to spend extra effort to tune $\kappa_i$) and that there is a potential theory-practice gap as shown by Figure 4.

7. This paper compares with [Mitra et al. 2022] in Appendix C.2. Is it true that [Mitra et al. 2022]'s regret bounds are problem-independent bounds? If so, it is unfair to say their bounds are worse than the problem-dependent bounds in this work.

8. Typos:

   8.1. Citations should be in parentheses

   8.2. Line 3 in Algorithm 1: $n_{ii, k}(t) \rightarrow n_{i, k}(t)$

   8.3. In Equation (2): $\% \rightarrow \emptyset$

   8.4. In proof of Theorem 2, the first equation on page 20, $n_{i,k}(t) \rightarrow n_{i, k}(T)$

    8.5. In proof of Lemma 2, in the last sentence, $$|\mathcal{B}_{i,k}(t) \cap \mathcal{H}|+|\mathcal{B}_{i,k}(t) \cap \mathcal{H}|\rightarrow |\mathcal{B}_{i,k}(t) \cap \mathcal{H}|+|\mathcal{B}_{i,k}(t) \cap \mathcal{F}|$$

**Strengths And Weaknesses:**

### Strengths

1. The cooperative bandits with the Byzantine agents model is novel and interesting.
2. The trim mean algorithmic technique is simple to understand and effective. The proposed message filter algorithm is straightforward and intuitive and possibly can be combined with other bandit algorithms. But there are some confusions that should be clarified as stated in the request for changes section.
3. The simulation results are sufficient.

### Weaknesses

1. The consensus-based communication policy is not widely known in the community. The authors should explain the setting better (see the first bullet for Requested Changes).
2. The theoretical results do not support the statement in the paper. See the second bullet below.
3.

---

> ### Author Response · Authors · 2024-06-22
> **Response to Reviewer 2sdX**
>
> We thank the reviewer for positive comments and valuable suggestions. We have corrected the typos and added a discussion for limitation in section Conclusion. The other response can be summarized as follows.
>
> **Discussion on the consistency filter:** The two filters, as a whole, serve as a preparation for each agent to formulate an accurate reward mean estimate so as to find the optimal arm. The trimmed-mean filter, as the reviewer mentioned, is for resilience purpose, as it effectively remove the influence from the Byzantine agents. However, it is important to note that the Byzantine influence is not the only factor that can affect the accuracy of the reward mean estimate. The reward information from a normal neighbor may also be inaccurate if the neighbor has insufficient samplings on the arm. And, if such inaccurate information is used by the agent in the update of the reward mean estimate, it may decrease the accuracy of the reward mean estimate and slow down the learning process. Our consistency filter is thus designed to avoid the situation and speed up the learning process. This point is also discussed in Section 3.
>
> **Theoretical results:** We have removed the redundant $\log T$ in the RHS of the equation in Theorem 1. Theorem 1 itself does not require any additional assumptions. We do mention that if certain requirement is satisfied, the lower bound is better than that in the single-agent scenario though, but it is just a supplementary property. We have made a minor change in the statement to make it less confusing.
>
> The reviewer did raise a good point, since theoretically, both our setting and the conventional single-agent setting can only provide an upper bound for agents' regret instead of computing the accurate regret, by saying better performance, we only mean ``better regret upper bound'', we have made corresponding adjustment to the statement. Notably, our algorithm does show better performance in experiments.
>
> **Proof of Theorem 3:** We have corrected the typo in eq. (17).
>
> **Gap between upper and lower bounds:** Firstly, despite the extensive history of bandit studies, even within the traditional single-agent framework, there remains a persistent gap between the upper and lower bounds. This discrepancy is likely inherent to bandit problems. Moving towards our resilient setting, the existence of the Byzantine agents can largely degrade the normal agents' performance. However, it is extremely difficult to analyze an algorithm-independent ``minimax'' affect the Byzantine agents can bring to the regret of the normal agents, thus in the analysis of the lower bound, we are only able to provide a relatively loose bound that does not take the Byzantine affect into account. That is why the upper and lower bounds do not match.
>
> **Discussion in Appendix E:** We have changed the notation used in this appendix section to $\tilde z$. The comparison is made between our proposed algorithm and a modified version of [Zhu \& Liu (2021)] to show that even with a filtering process, the updating policy used in the existing literature cannot solve the resilient bandit problem. Due to this purpose, we modify the algorithm in  [Zhu \& Liu (2021)] (the algorithm is designed for a classic bandit problem without considering Byzantine attack) by adding a filtering process (without a filtering process, the algorithm clearly cannot deal with the resilient problem due to the Byzantine affect). The updating equation used in the comparison is given at the end of page 17 instead of the first equation in Appendix E.
>
> **Comparison with [Mitra et al. 2022]:** [Mitra et al. 2022] and our paper have different focuses and restrictions. [Mitra et al. 2022] studies a linear bandit problem, which has a more general reward assumption compared to the stochastic bandit setting studied in our paper. On the other hand,
> we study on a more general graph/communication scenario, as
> [Mitra et al. 2022] assumes there is a central agent that has access to all the information in the network, which is a restriction in communication compared to our decentralized communicating scenario. The scenarios both work study do not contain each other but have overlapping situations, in which our work provides a better bound.

---

> > ### Comment · Reviewer_2sdX · 2024-06-25
> > **Thanks for your response**
> >
> > A few more clarification comments:
> >
> > Regarding theoretical results: Can the RHS of Theorem 6 also be improved?
> >
> > Appendix E: I do not see revision in Appendix E. Also, I would suggest the authors introduce the algorithm of [Zhu & Liu (2021)] and this paper’s modification in detail to make this paper self-contained.
> >
> > Comparison with [Mitra et al. 2022]: I would suggest the authors to explicitly explain which “overlapping situation” they actually mean in Appendix C2 to make it a fair comparison.

---

> > > ### Author Response · Authors · 2024-06-27
> > > **Response to Reviewer 2sdX**
> > >
> > > Thank you for your suggestions, we have made the corresponding revisions accordingly.

---

### Decision · Action_Editor_2DSF · 2024-07-14

**Recommendation:** Accept as is

**Comment:**

Reviewer 2sdX in particular found some minor errors, which the authors have already addressed. The other reviewers had minor comments and questions which encouraged the authors to make some nice changes to the manuscript, including an additional simulation. This paper easily satisfies both criteria for acceptance, so I am happy to recommend acceptance.

**Audience:**

The paper studies the interesting setting of decentralised UCB, where some agents can send arbitrarily wrong information in the mean or confidence of the reward. All three reviewers agree the paper is of interest to the TMLR community. So much so, that two recommended that the paper would be appropriate for presentation at ICLR.

**Claims And Evidence:**

All reviewers agree that claims are accurate and supported by evidence or appropriate proofs. Several minor errors were raised during the review/rebuttal stage, and the authors have made appropriate changes to the paper.